# LEARNING WITH STOCHASTIC ORDERS

**Carles Domingo-Enrich**
New York University
cd2754@nyu.edu

**Yair Schiff**[*]
Cornell University
yzs2@cornell.edu

**Youssef Mroueh**
IBM Research AI
mroueh@us.ibm.com

## ABSTRACT

Learning high-dimensional distributions is often done with explicit likelihood modeling or implicit modeling via minimizing integral probability metrics (IPMs). In this paper, we expand this learning paradigm to stochastic orders, namely, the *convex* or *Choquet order* between probability measures. Towards this end, exploiting the relation between convex orders and optimal transport, we introduce the Choquet-Toland distance between probability measures, that can be used as a drop-in replacement for IPMs. We also introduce the *Variational Dominance Criterion* (VDC) to learn probability measures with dominance constraints, that encode the desired stochastic order between the learned measure and a known baseline. We analyze both quantities and show that they suffer from the curse of dimensionality and propose surrogates via input convex maxout networks (ICMNs), that enjoy parametric rates. We provide a min-max framework for learning with stochastic orders and validate it experimentally on synthetic and high-dimensional image generation, with promising results. Finally, our ICMNs class of convex functions and its derived Rademacher Complexity are of independent interest beyond their application in convex orders. Code to reproduce experimental results is available here.

## 1 INTRODUCTION

Learning complex high-dimensional distributions with implicit generative models (Goodfellow et al., 2014; Mohamed & Lakshminarayanan, 2017; Arjovsky et al., 2017) via minimizing *integral probability metrics* (IPMs) (Müller, 1997a) has led to the state of the art generation across many data modalities (Karras et al., 2019; De Cao & Kipf, 2018; Padhi et al., 2020). An IPM compares probability distributions with a witness function belonging to a function class $\mathcal{F}$, e.g., the class of Lipchitz functions, which makes the IPM correspond to the Wasserstein distance 1. While estimating the witness function in such large function classes suffers from the curse of dimensionality, restricting it to a class of neural networks leads to the so called neural net distance (Arora et al., 2017) that enjoys parametric statistical rates.

In probability theory, the question of comparing distributions is not limited to assessing only equality between two distributions. Stochastic orders were introduced to capture the notion of dominance between measures. Similar to IPMs, stochastic orders can be defined by looking at the integrals of measures over function classes $\mathcal{F}$ (Müller, 1997b). Namely, for $\mu_+, \mu_- \in \mathcal{P}_1(\mathbb{R}^d)$, $\mu_+$ dominates $\mu_-$, or $\mu_- \preceq \mu_+$, if for any function $f \in \mathcal{F}$, we have $\int_{\mathbb{R}^d} f(x)\,d\mu_-(x) \leq \int_{\mathbb{R}^d} f(x)\,d\mu_+(x)$ (See Figure 1a for an example). In the present work, we focus on the *Choquet* or *convex* order (Ekeland & Schachermayer, 2014) generated by the space of convex functions (see Sec. 2 for more details).

Previous work has focused on learning with stochastic orders in the one dimensional setting, as it has prominent applications in mathematical finance and distributional reinforcement learning (RL). The survival function gives a characterization of the convex order in one dimension (See Figure 1b and Sec. 2 for more details). For instance, in portfolio optimization (Xue et al., 2020; Post et al., 2018; Dentcheva & Ruszczynski, 2003) the goal is to find the portfolio that maximizes the expected return under dominance constraints between the return distribution and a benchmark distribution.

---

[*]Work done while at IBM.

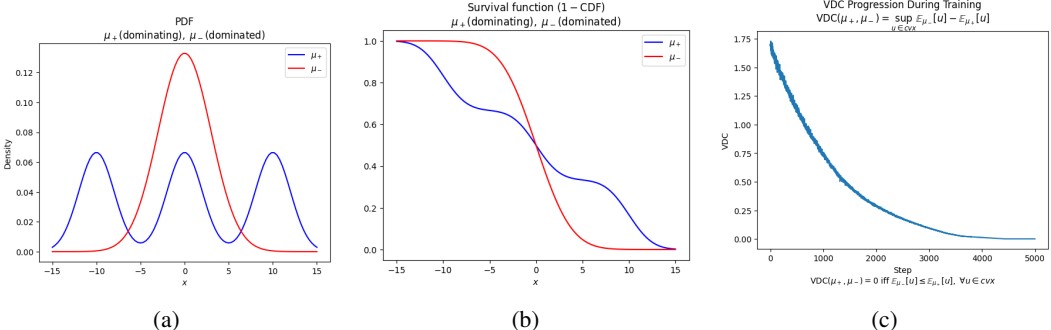

Figure 1: VDC example in 1D. Figure 1a :$\mu_+$ is mixture of 3 Gaussians , $\mu_-$ corresponds to a single mode of the mixture. $\mu_+$ dominates $\mu_-$ in the convex order. Figure 1b : uni-variate characterization of the convex order with survival functions (See Sec. 2 for details). Figure 1c: Surrogate VDC computation with Input Convex Maxout Network and gradient descent. The surrogate VDC tends to zero at the end of the training and hence characterizes the convex dominance of $\mu_+$ on $\mu_-$.

A similar concept was introduced in distributional RL (Martin et al., 2020) for learning policies with dominance constraints on the distribution of the reward. While these works are limited to the univariate setting, our work is the first, to the best of our knowledge, that provides a computationally tractable characterization of stochastic orders that is sample efficient and scalable to high dimensions.

The paper is organized as follows: in Sec. 3 we introduce the *Variational Dominance Criterion* (VDC); the VDC between measures $\mu_+$ and $\mu_-$ takes value 0 if and only if $\mu_+$ dominates $\mu_-$ in the convex order, but it suffers from the curse of dimension and cannot be estimated efficiently from samples. To remediate this, in Sec. 4 we introduce a VDC surrogate via Input Convex Maxout Networks (ICMNs). ICMNs are new variants of Input Convex Neural Nets (Amos et al., 2017) that we propose as proxy for convex functions and study their complexity. We show in Sec. 4 that the surrogate VDC has parametric rates and can be efficiently estimated from samples. The surrogate VDC can be computed using (stochastic) gradient descent on the parameters of the ICMN and can characterize convex dominance (See Figure 1c). We then show in Sec. 5 how to use the VDC and its surrogate to define a pseudo-distance on the probability space. Finally, in Sec. 6 we propose penalizing generative models training losses with the surrogate VDC to learn implicit generative models that have better coverage and spread than known baselines. This leads to a min-max game similar to GANs. We validate our framework in Sec. 7 with experiments on portfolio optimization and image generation.

## 2 THE CHOQUET OR CONVEX ORDER

Denote by $\mathcal{P}(\mathbb{R}^d)$ the set of Borel probability measures on $\mathbb{R}^d$ and by $\mathcal{P}_1(\mathbb{R}^d) \subset \mathcal{P}(\mathbb{R}^d)$ the subset of those which have finite first moment: $\mu \in \mathcal{P}_1(\mathbb{R})$ if and only if $\int_{\mathbb{R}^d} \|x\| \, d\mu(x) < +\infty$.

**Comparing probability distributions** Integral probability metrics (IPMs) are pseudo-distances between probability measures $\mu, \nu$ defined as $d_{\mathcal{F}}(\mu, \nu) = \sup_{f \in \mathcal{F}} \mathbb{E}_\mu f - \mathbb{E}_\nu f$, for a given function class $\mathcal{F}$ which is symmetric with respect to sign flips. They are ubiquitous in optimal transport and generative modeling to compare distributions: if $\mathcal{F}$ is the set of functions with Lipschitz constant 1, then the resulting IPM is the 1-Wasserstein distance; if $\mathcal{F}$ is the unit ball of an RKHS, the IPM is its maximum mean discrepancy. Clearly, $d_{\mathcal{F}}(\mu, \nu) = 0$ if and only $\mathbb{E}_\mu f = \mathbb{E}_\nu f$ for all $f \in \mathcal{F}$, and when $\mathcal{F}$ is large enough, this is equivalent to $\mu = \nu$.

**The Choquet or convex order** When the class $\mathcal{F}$ is not symmetric with respect to sign flips, comparing the expectations $\mathbb{E}_\mu f$ and $\mathbb{E}_\nu f$ for $f \in \mathcal{F}$ does not yield a pseudo-distance. In the case where $\mathcal{F}$ is the set of convex functions, the convex order naturally arises instead:

**Definition 1** (Choquet order, Ekeland & Schachermayer (2014), Def. 4). *For $\mu_-, \mu_+ \in \mathcal{P}_1(\mathbb{R}^d)$, we say that $\mu_- \preceq \mu_+$ if for any convex function $f : \mathbb{R}^d \to \mathbb{R}$, we have*

$$\int_{\mathbb{R}^d} f(x) \, d\mu_-(x) \leq \int_{\mathbb{R}^d} f(x) \, d\mu_+(x).$$

$\mu_- \preceq \mu_+$ is classically denoted as "$\mu_-$ is a balayée of $\mu_+$", or "$\mu_+$ dominates $\mu_-$". It turns out that $\preceq$ is a partial order on $\mathcal{P}_1(\mathbb{R}^d)$, meaning that reflexivity ($\mu \preceq \mu$), antisymmetry (if $\mu \preceq \nu$ and $\nu \preceq \mu$, then $\mu = \nu$), and transitivity (if $\mu_1 \preceq \mu_2$ and $\mu_2 \preceq \mu_3$, then $\mu_1 \preceq \mu_3$) hold. As an example, if $\mu_-$, $\mu_+$ are Gaussians $\mu_- = N(0, \Sigma_-)$, $\mu_+ = N(0, \Sigma_+)$, then $\mu_- \preceq \mu_+$ if and only if $\Sigma_- \preceq \Sigma_+$ in the positive-semidefinite order (Müller, 2001). Also, since linear functions are convex, $\mu_- \preceq \mu_+$ implies that both measures have the same expectation: $\mathbb{E}_{x \sim \mu_-} x = \mathbb{E}_{x \sim \mu_+} x$.

In the univariate case, $\mu_- \preceq \mu_+$ implies that $\mathrm{supp}(\mu_-) \subseteq \mathrm{supp}(\mu_+)$ and that $\mathrm{Var}(\mu_-) \leq \mathrm{Var}(\mu_+)$, and we have that $\mu_- \preceq \mu_+$ holds if and only if for all $x \in \mathbb{R}$, $\int_x^{+\infty} \bar{F}_{\mu_-}(t)\, dt \leq \int_x^{+\infty} \bar{F}_{\mu_+}(t)\, dt$, where $\bar{F}$. is the survival function (one minus the cumulative distribution function). Note that this characterization can be checked efficiently if one has access to samples of $\mu_-$ and $\mu_+$.

In the high-dimensional case, there exists an alternative characterization of the convex order:

**Proposition 1** (Ekeland & Schachermayer (2014), Thm. 10). *If $\mu_-, \mu_+ \in \mathcal{P}_1(\mathbb{R}^d)$, we have $\mu_- \preceq \mu_+$ if and only if there exists a Markov kernel $R$ (i.e. $\forall x \in \mathbb{R}^d$, $\int_{\mathbb{R}^d} y\, dR_x(y) = x$) such that $\mu_+ = \int_{\mathbb{R}^d} R_x\, d\mu_-$.*

Equivalently, there exists a coupling $(X, Y)$ such that $\mathrm{Law}(X) = \mu_-$, $\mathrm{Law}(Y) = \mu_+$ and $X = \mathbb{E}(Y|X)$ almost surely. Intuitively, this means that $\mu_+$ is more *spread out* than $\mu_-$. Remark that this characterization is difficult to check, especially in high dimensions.

## 3 THE VARIATIONAL DOMINANCE CRITERION

In this section, we present a quantitative way to deal with convex orders. Given a bounded open convex subset $\Omega \subseteq \mathbb{R}^d$ and a compact set $\mathcal{K} \subseteq \mathbb{R}^d$, let $\mathcal{A} = \{u : \Omega \to \mathbb{R},\ u \text{ convex and } \nabla u \in \mathcal{K} \text{ almost everywhere}\}$. We define the *Variational Dominance Criterion* (VDC) between probability measures $\mu_+$ and $\mu_-$ supported on $\Omega$ analogously to IPMs, replacing $\mathcal{F}$ by $\mathcal{A}$:

$$\mathrm{VDC}_{\mathcal{A}}(\mu_+ \| \mu_-) := \sup_{u \in \mathcal{A}} \int_{\Omega} u\, d(\mu_- - \mu_+). \tag{1}$$

Remark that when $0 \in \mathcal{K}$, $\mathrm{VDC}_{\mathcal{A}}(\mu_+ \| \mu_-) \geq 0$ because the zero function belongs to the set $\mathcal{A}$. We reemphasize that since $\mathcal{A}$ is not symmetric with respect to sign flips as $f \in \mathcal{A}$ does not imply $-f \in \mathcal{A}$, the properties of the VDC are very different from those of IPMs. Most importantly, the following proposition, shown in App. A, links the VDC to the Choquet order.

**Proposition 2.** *Let $\mathcal{K}$ compact such that the origin belongs to the interior of $\mathcal{K}$. If $\mu_+, \mu_- \in \mathcal{P}(\Omega)$, $\mathrm{VDC}_{\mathcal{A}}(\mu_+ \| \mu_-) := \sup_{u \in \mathcal{A}} \int_{\Omega} u\, d(\mu_- - \mu_+) = 0$ if and only if $\int_{\Omega} u\, d(\mu_- - \mu_+) \leq 0$ for any convex function on $\Omega$ (i.e. $\mu_- \preceq \mu_+$ according to the Choquet order).*

That is, Proposition 2 states that the VDC between $\mu_+$ and $\mu_-$ takes value 0 if and only if $\mu_+$ dominates $\mu_-$. Combining this with the interpretation of Proposition 1, we see that intuitively, *the quantity $\mathrm{VDC}_{\mathcal{A}}(\mu_+ \| \mu_-)$ is small when $\mu_+$ is more spread out than $\mu_-$, and large otherwise*. Hence, if we want to enforce or induce a Choquet ordering between two measures in an optimization problem, we can include the VDC (or rather, its surrogate introduced in Sec. 4) as a penalization term in the objective. Before this, we explore the connections between VDC and optimal transport, and study some statistical properties of the VDC.

### 3.1 THE VDC AND OPTIMAL TRANSPORT

Toland duality provides a way to interpret the VDC through the lens of optimal transport. In the following, $W_2(\mu, \nu)$ denotes the 2-Wasserstein distance between $\mu$ and $\nu$.

**Theorem 1** (Toland duality, adapted from Thm. 1 of Carlier (2008)). *For any $\mu_+, \mu_- \in \mathcal{P}(\Omega)$, the VDC satisfies:*

$$\mathrm{VDC}_{\mathcal{A}}(\mu_+ \| \mu_-) = \sup_{\nu \in \mathcal{P}(\mathcal{K})} \left\{ \frac{1}{2} W_2^2(\mu_+, \nu) - \frac{1}{2} W_2^2(\mu_-, \nu) \right\} - \frac{1}{2} \int_{\Omega} \|x\|^2\, d(\mu_+ - \mu_-)(x) \tag{2}$$

*The optimal convex function $u$ of VDC in (1) and the optimal $\nu$ in the right-hand side of (2) satisfy $(\nabla u)_{\#} \mu_+ = (\nabla u)_{\#} \mu_- = \nu$, where $(\nabla u)_{\#} \mu_+$ denotes the pushforward of $\mu_+$ by $\nabla u$.*

Note that under the assumption $0 \in \mathcal{K}$, Theorem 1 implies that $\text{VDC}_{\mathcal{A}}(\mu_+ \| \mu_-) = 0$ if and only if $W_2^2(\mu_+, \nu) - \frac{1}{2} \int_{\Omega} \|x\|^2 \, d\mu_+ \leq W_2^2(\mu_-, \nu) - \frac{1}{2} \int_{\Omega} \|x\|^2 \, d\mu_-$ for any $\nu \in \mathcal{P}(\mathcal{K})$. Under the equivalence $\text{VDC}_{\mathcal{A}}(\mu_+ \| \mu_-) = 0 \iff \mu_+ \succeq \mu_-$ shown by Proposition 2, this provides yet another characterization of the convex order for arbitrary dimension.

## 3.2 STATISTICAL RATES FOR VDC ESTIMATION

In this subsection, we present an upper bound on the statistical rate of estimation of $\text{VDC}_{\mathcal{A}}(\mu \| \nu)$ using the estimator $\text{VDC}_{\mathcal{A}}(\mu_n \| \nu_n)$ based on the empirical distributions $\mu_n = \frac{1}{n} \sum_{i=1}^n \delta_{x_i}$, $\nu_n = \frac{1}{n} \sum_{i=1}^n \delta_{y_i}$ built from i.i.d. samples $(x_i)_{i=1}^n$, $(y_i)_{i=1}^n$ from $\mu$ and $\nu$, respectively.

**Theorem 2.** *Let $\Omega = [-1, 1]^d$ and $\mathcal{K} = \{x \in \mathbb{R}^d \mid \|x\|_2 \leq C\}$ for an arbitrary $C > 0$. With probability at least $1 - \delta$,*

$$|\text{VDC}_{\mathcal{A}}(\mu \| \nu) - \text{VDC}_{\mathcal{A}}(\mu_n \| \nu_n)| \leq \sqrt{18 C^2 d \log(\tfrac{\delta}{4})} (\tfrac{2}{\sqrt{n}}) + 8 K n^{-\frac{2}{d}},$$

*where $K$ depends on $C$ and $d$.*

The proof of this result is in App. A. The dependency on $n^{-\frac{2}{d}}$ is indicative of the curse of dimension: we need a number of samples $n$ exponential in $d$ to control the estimation error. While Theorem 2 only shows an upper bound on the difference between the VDC and its estimator, in Subsec. 5.1 we study a related setting where a $\tilde{\Omega}(n^{-\frac{2}{d}})$ lower bound is available. Hence, we hypothesize that VDC estimation is in fact cursed by dimension in general.

## 4 A VDC SURROGATE VIA INPUT CONVEX MAXOUT NETWORKS

Given the link between the VDC and the convex order, one is inclined to use the VDC as a quantitative proxy to induce convex order domination in optimization problems. Estimating the VDC implies solving an optimization problem over convex functions. In practice, we only have access to the empirical versions $\mu_n, \nu_n$ of the probability measures $\mu, \nu$; we could compute the VDC between the empirical measures by solving a linear program similar to the ones used in non-parametric convex regression (Hildreth, 1954). However, the statistical rates for the VDC estimation from samples are cursed by dimension (Subsec. 3.2), which means that we would need a number of samples exponential in the dimension to get a good estimate. Our approach is to focus on a surrogate problem instead:

$$\sup_{u \in \hat{\mathcal{A}}} \int_{\Omega} u \, d(\mu_- - \mu_+),$$

where $\hat{\mathcal{A}}$ is a class of neural network functions included in $\mathcal{A}$ over which we can optimize efficiently. In constructing $\hat{\mathcal{A}}$, we want to hardcode the constraints $u$ *convex* and $\nabla u \in \mathcal{K}$ *almost everywhere* into the neural network architectures. A possible approach would be to use the input convex neural networks (ICNNs) introduced by Amos et al. (2017), which have been used as a surrogate of convex functions for generative modeling with normalizing flows (Huang et al., 2021) in optimal transport (Korotin et al., 2021a;b; Huang et al., 2021; Makkuva et al., 2020) and large-scale Wasserstein flows (Alvarez-Melis et al., 2021; Bunne et al., 2021; Mokrov et al., 2021).

However, we found in early experimentation that a superior alternative is to use input convex maxout networks (ICMNs), which are maxout networks (Goodfellow et al., 2013) that are convex with respect to inputs. Maxout networks and ICMNs are defined as follows:

**Definition 2** (Maxout networks). *For a depth $L \geq 2$, let $\mathcal{M} = (m_1, \ldots, m_L)$ be a vector of positive integers such that $m_1 = d$. Let $F_{L, \mathcal{M}, k}$ be the space of $k$-maxout networks of depth $L$ and widths $\mathcal{M}$, which contains functions of the form*

$$f(x) = \tfrac{1}{\sqrt{m_L}} \sum_{i=1}^{m_L} a_i \max_{j \in [k]} \langle w_{i,j}^{(L-1)}, (x^{(L-1)}, 1) \rangle, \qquad a_i \in \mathbb{R}, \; w_{i,j}^{(L-1)} \in \mathbb{R}^{m_{L-1}+1} \quad (3)$$

*where for any $2 \leq \ell \leq L - 1$, and any $1 \leq i \leq m_\ell$, the $i$-th component of $x^{(\ell)} = (x_1^{(\ell)}, \ldots, x_{m_\ell}^{(\ell)})$ is computed recursively as:*

$$x_i^{(\ell)} = \tfrac{1}{\sqrt{m_\ell}} \max_{j \in [k]} \langle w_{i,j}^{(\ell-1)}, (x^{(\ell-1)}, 1) \rangle, \qquad w_{i,j}^{(\ell)} \in \mathbb{R}^{m_\ell+1}, \quad (4)$$

*with $x^{(1)} = x$.*

**Definition 3** (Input convex maxout networks or ICMNs). *A maxout network $f$ of the form* (3)-(4) *is an input convex maxout network if (i) for any $1 \leq i \leq M_L$, $a_i \geq 0$, and (ii) for any $2 < \ell \leq L - 1$, $1 \leq i \leq m_{\ell+1}$, $1 \leq j \leq k$, the first $m_\ell$ components of $w_{i,j}^{(\ell)}$ are non-negative. We denote the space of ICMNs as $F_{L,\mathcal{M},k,+}$.*

In other words, a maxout network is an ICMN if all the non-bias weights beyond the first layer are constrained to be non-negative. This definition is analogous to the one of ICNNs in Amos et al. (2017), which are also defined as neural networks with positivity constraints on non-bias weights beyond the first layer. Proposition 5 in App. B shows that ICMNs are convex w.r.t to their inputs.

It remains to impose the condition $\nabla u \in \mathcal{K}$ *almost everywhere*, which in practice is enforced by adding the norms of the weights as a regularization term to the loss function. For theoretical purposes, we define $F_{L,\mathcal{M},k}(1)$ (resp. $F_{L,\mathcal{M},k,+}(1)$) as the subset of $F_{L,\mathcal{M},k}$ (resp. $F_{L,\mathcal{M},k,+}$) such that for all $1 \leq \ell \leq L - 1$, $1 \leq i \leq m_\ell$, $1 \leq j \leq k$, $\|w_{i,j}^{(\ell)}\|_2 \leq 1$, and $\|a\|_2 = \sum_{i=1}^{m_L} a_i^2 \leq 1$. The following proposition, proven in App. B, shows simple bounds on the values of the functions in $F_{L,\mathcal{M},k}(1)$ and their derivatives.

**Proposition 3.** *Let $f$ be an ICMN that belongs to $F_{L,\mathcal{M},k}(1)$. For $x$ almost everywhere in $\mathcal{B}_1(\mathbb{R}^d)$, $\|\nabla f(x)\| \leq 1$. Moreover, for all $x \in \mathcal{B}_1(\mathbb{R}^d)$, $|f(x)| \leq L$, and for $1 \leq \ell \leq L$, $\|x^{(\ell)}\| \leq \ell$.*

When $\mathcal{K} = \mathcal{B}_1(\mathbb{R}^d)$, we have that the space of ICMNs $F_{L,\mathcal{M},k,+}(1)$ is included in $\mathcal{A}$. Hence, we define the surrogate VDC associated to $F_{L,\mathcal{M},k,+}(1)$ as:

$$\text{VDC}_{F_{L,\mathcal{M},k,+}(1)}(\mu_+||\mu_-) = \sup_{u \in F_{L,\mathcal{M},k,+}(1)} \int_\Omega u \, d(\mu_- - \mu_+). \tag{5}$$

**Theorem 3.** *Suppose that for all $1 \leq \ell \leq L$, the widths $m_\ell$ satisfy $m_\ell \leq m$, and assume that $\mu, \nu$ are supported on the ball of $\mathbb{R}^d$ of radius $r$. With probability at least $1 - \delta$,*

$$|\text{VDC}_{F_{L,\mathcal{M},k,+}(1)}(\mu||\nu) - \text{VDC}_{F_{L,\mathcal{M},k,+}(1)}(\mu_n||\nu_n)|$$
$$\leq \sqrt{18r^2 \log(\tfrac{\delta}{4})}\left(\tfrac{2}{\sqrt{n}}\right) + 512\sqrt{\tfrac{(L-1)km(m+1)}{n}}\left(\sqrt{(L+1)\log(2)} + \tfrac{\log(L^2+1)}{2} + \tfrac{\sqrt{\pi}}{2}\right),$$

We see from Theorem 3 that $\text{VDC}_{F_{L,\mathcal{M},k,+}(1)}$ in contrast to $\text{VDC}_\mathcal{A}$ has parametric rates and hence favorable properties to be estimated from samples. In the following section, wes see that VDC can be used to defined a pseudo-distance on the probability space.

## 5 FROM THE CONVEX ORDER BACK TO A PSEUDO-DISTANCE

We define the *Choquet-Toland distance* (CT distance) as the map $d_{\text{CT},\mathcal{A}} : \mathcal{P}(\Omega) \times \mathcal{P}(\Omega) \to \mathbb{R}$ given by

$$d_{\text{CT},\mathcal{A}}(\mu_+, \mu_-) := \text{VDC}_\mathcal{A}(\mu_+||\mu_-) + \text{VDC}_\mathcal{A}(\mu_-||\mu_+).$$

That is, the CT distance between $\mu_+$ and $\mu_-$ is simply the sum of Variational Dominance Criteria. Applying Theorem 1, we obtain that $d_{\text{CT},\mathcal{A}}(\mu_+, \mu_-) = \frac{1}{2}(\sup_{\nu \in \mathcal{P}(\mathcal{K})} \{W_2^2(\mu_+, \nu) - W_2^2(\mu_-, \nu)\} + \sup_{\nu \in \mathcal{P}(\mathcal{K})} \{W_2^2(\mu_+, \nu) - W_2^2(\mu_-, \nu)\})$. The following result, shown in App. C, states that $d_{CT,K}$ is indeed a distance.

**Theorem 4.** *Suppose that the origin belongs to the interior of $\mathcal{K}$. $d_{CT,\mathcal{A}}$ is a distance, i.e. it fulfills*

*(i) $d_{CT,\mathcal{A}}(\mu_+, \mu_-) \geq 0$ for any $\mu_+, \mu_- \in \mathcal{P}(\Omega)$ (non-negativity).*

*(ii) $d_{CT,\mathcal{A}}(\mu_+, \mu_-) = 0$ if and only if $\mu_+ = \mu_-$ (identity of indiscernibles).*

*(iii) If $\mu_1, \mu_2, \mu_3 \in \mathcal{P}(\Omega)$, we have that $d_{CT,\mathcal{A}}(\mu_1, \mu_2) \leq d_{CT,\mathcal{A}}(\mu_1, \mu_3) + d_{CT,\mathcal{A}}(\mu_3, \mu_2)$.*

As in (5), we define the surrogate CT distance as:

$$d_{\text{CT},F_{L,\mathcal{M},k,+}(1)}(\mu_+, \mu_-) = \text{VDC}_{F_{L,\mathcal{M},k,+}(1)}(\mu_+||\mu_-) + \text{VDC}_{F_{L,\mathcal{M},k,+}(1)}(\mu_-||\mu_+). \tag{6}$$

## 5.1 Statistical rates for CT distance estimation

We show almost-tight upper and lower bounds on the expectation of the Choquet-Toland distance between a probability measure and its empirical version. Namely, $\mathbb{E}[d_{\mathrm{CT},\mathcal{A}}(\mu, \mu_n)] = O(n^{-\frac{2}{d}}), \Omega(n^{-\frac{2}{d}}/\log(n))$.

**Theorem 5.** *Let $C_0, C_1$ be universal constants independent of the dimension $d$. Let $\Omega = [-1, 1]^d$, and $\mathcal{K} = \{x \in \mathbb{R}^d \mid \|x\|_2 \leq C\}$. Let $\mu_n$ be the $n$-sample empirical measure corresponding to a probability measure $\mu$ over $[-1, 1]^d$. When $\mu$ is the uniform probability measure and $n \geq C_1 \log(d)$, we have that*

$$\mathbb{E}[d_{CT,\mathcal{A}}(\mu, \mu_n)] \geq C_0 \sqrt{\tfrac{C}{d(1+C)\log(n)}} n^{-\frac{2}{d}}. \tag{7}$$

*For any probability measure $\mu$ over $[-1, 1]^d$,*

$$\mathbb{E}[d_{CT,\mathcal{A}}(\mu, \mu_n)] \leq K n^{-\frac{2}{d}}, \tag{8}$$

*where $K$ is a constant depending on $C$ and $d$ (but not the measure $\mu$).*

Overlooking logarithmic factors, we can summarize Theorem 5 as $\mathbb{E}[d_{\mathrm{CT},\mathcal{A}}(\mu, \mu_n)] \asymp n^{-\frac{2}{d}}$. The estimation of the CT distance is cursed by dimension: one needs a sample size exponential with the dimension $d$ for $\mu_n$ to be at a desired distance from $\mu$. It is also interesting to contrast the rate $n^{-\frac{2}{d}}$ with the rates for similar distances. For example, for the $r$-Wasserstein distance we have $\mathbb{E}[W_r^r(\mu, \mu_n)] \asymp n^{-\frac{r}{d}}$ (Singh & Póczos (2018), Section 4). Given the link of the CT distance and VDC with the squared 2-Wasserstein distance (see Subsec. 3.1), the $n^{-\frac{2}{d}}$ rate is natural.

The proof of Theorem 5, which can be found in App. D, is based on upper-bounding and lower-bounding the metric entropy of the class of bounded Lipschitz convex functions with respect to an appropriate pseudo-norm. Then we use Dudley's integral bound and Sudakov minoration to upper and lower-bound the Rademacher complexity of this class, and we finally show upper and lower bounds of the CT distance by the Rademacher complexity. Bounds on the metric entropy of bounded Lipschitz convex functions have been computed and used before (Balazs et al., 2015; Guntuboyina & Sen, 2013), but in $L^p$ and supremum norms, not in our pseudo-norm.

Next, we see that the surrogate CT distance defined in (6) does enjoy parametric estimation rates.

**Theorem 6.** *Suppose that for all $1 \leq \ell \leq L$, the widths $m_\ell$ satisfy $m_\ell \leq m$. We have that*

$$\mathbb{E}[d_{CT,F_{L,\mathcal{M},k,+}(1)}(\mu, \mu_n)] \leq 256 \sqrt{\tfrac{(L-1)km(m+1)}{n}} \left( \sqrt{(L+1)\log(2)} + \tfrac{\log(L^2+1)}{2} + \tfrac{\sqrt{\pi}}{2} \right),$$

In short, we have that $\mathbb{E}[d_{\mathrm{CT},F_{L,\mathcal{M},k,+}(1)}(\mu, \mu_n)] = O(Lm\sqrt{\tfrac{k}{n}})$. Hence, if we take the number of samples $n$ larger than $k$ times the squared product of width and depth, we can make the surrogate CT distance small. Theorem 6, proven in App. D, is based on a Rademacher complexity bound for the space $F_{L,\mathcal{M},k}(1)$ of maxout networks which may be of independent interest; to our knowledge, existing Rademacher complexity bounds for maxout networks are restricted to depth-two networks (Balazs et al., 2015; Kontorovich, 2018).

Theorems 5 and 6 show the advantages of the surrogate CT distance over the CT distance are not only computational but also statistical; the CT distance is such a strong metric that moderate-size empirical versions of a distribution are always very far from it. Hence, it is not a good criterion to compare how close an empirical distribution is to a population distribution. In contrast, the surrogate CT distance between a distribution and its empirical version is small for samples of moderate size. An analogous observation for the Wasserstein distance versus the neural net distance was made by Arora et al. (2017).

If $\mu_n, \nu_n$ are empirical versions of $\mu, \nu$, it is also interesting to bound $|d_{\mathrm{CT},F_{L,\mathcal{M},k,+}(1)}(\mu_n, \nu_n) - d_{\mathrm{CT},\mathcal{A}}(\mu, \nu)| \leq |d_{\mathrm{CT},F_{L,\mathcal{M},k,+}(1)}(\mu_n, \nu_n) - d_{\mathrm{CT},F_{L,\mathcal{M},k,+}(1)}(\mu, \nu)| + |d_{\mathrm{CT},F_{L,\mathcal{M},k,+}(1)}(\mu, \nu) - d_{\mathrm{CT},\mathcal{A}}(\mu, \nu)|$. The first term has a $O(\sqrt{k/n})$ bound following from Theorem 6, while the second term is upper-bounded by $2 \sup_{f \in \mathcal{A}} \inf_{\tilde{f} \in F_{L,\mathcal{M},k}(1)} \|f - \tilde{f}\|_\infty$, which is (twice) the approximation error

of the class $\mathcal{A}$ by the class $F_{L,\mathcal{M},k}(1)$. Such bounds have only been derived in $L = 2$: Balazs et al. (2015) shows that $\sup_{f \in \mathcal{A}} \inf_{\tilde{f} \in F_{2,(d,1),k}(1)} \|f - \tilde{f}\|_\infty = O(dk^{-2/d})$. Hence, we need $k$ exponential in $d$ to make the second term small, and thus $n \gg k$ exponential in $d$ to make the first term small.

# 6 LEARNING DISTRIBUTIONS WITH THE SURROGATE VDC AND CT DISTANCE

We provide a min-max framework to learn distributions with stochastic orders. As in the generative adversarial network (GAN, Goodfellow et al. (2014); Arjovsky et al. (2017)) framework, we parametrize probability measures implicitly as the pushforward $\mu = g_\#\mu_0$ of a base measure $\mu_0$ by a generator function $g$ in a parametric class $\mathcal{G}$ and we optimize over $g$. The loss functions involve a maximization over ICMNs corresponding to the computation of a surrogate VDC or CT distance (and possibly additional maximization problems), yielding a min-max problem analogous to GANs.

**Enforcing dominance constraints with the surrogate VDC.** In some applications, we want to optimize a loss $L : \mathcal{P}(\Omega) \to \mathbb{R}$ under the constraint that $\mu = g_\#\mu_0$ dominates a baseline measure $\nu$. We can enforce, or at least, bias $\mu$ towards the dominance constraint by adding a penalization term proportional to the surrogate VDC between $\mu$ and $\nu$, in application of Proposition 2.

A first instance of this approach appears in portfolio optimization (Xue et al., 2020; Post et al., 2018). Let $\xi = (\xi_1, \ldots, \xi_p)$ be a random vector of return rates of $p$ assets and let $Y_1 := G_1(\xi)$, $Y_2 := G_2(\xi)$ be real-valued functions of $\xi$ which represent the return rates of two different asset allocations or portfolios, e.g. $G_i(\xi) = \langle \omega_i, \xi \rangle$ with $\omega_i \in \mathbb{R}^p$. The goal is to find a portfolio $G_2$ that enhances a benchmark portfolio $G_1$ in a certain way. For a portfolio $G$ with return rate $Y := G(\xi)$, we let $F_Y^{(1)}(x) = P_\xi(Y \leq x)$ be the CDF of its return rate, and $F_Y^{(2)}(x) = \int_{-\infty}^x F_Y^{(1)}(x')\,dx'$. If $Y_1, Y_2$ are the return rates of $G_1, G_2$, we say that $Y_2$ dominates $Y_1$ in second order, or $Y_2 \succeq_2 Y_1$ if for all $x \in \mathbb{R}$, $F_{Y_2}^{(2)}(x) \leq F_{Y_1}^{(2)}(x)$, which intuitively means that the return rates $Y_2$ are less spread out than those $Y_1$, i.e. the risk is smaller. Formally, the portfolio optimization problem can be written as:

$$\max_{G_2} \mathbb{E}[Y_2 := G_2(\xi)], \qquad \text{s.t. } Y_2 \succeq_2 Y_1 := G_1(\xi). \tag{9}$$

It turns out that $Y_2 \succeq_2 Y_1$ if and only if $\mathbb{E}[(\eta - Y_2)_+] \leq \mathbb{E}[(\eta - Y_1)_+]$ for any $\eta \in \mathbb{R}$, or yet equivalently, if $\mathbb{E}[u(Y_2)] \geq \mathbb{E}[u(Y_1)]$ for all concave non-decreasing $u : \mathbb{R} \to \mathbb{R}$ (Dentcheva & Ruszczyński, 2004). Although different, note that the second order is intimately connected to the Choquet order for 1-dimensional distributions, and it can be handled with similar tools. Define $F_{L,\mathcal{M},k,-+}(1)$ as the subset of $F_{L,\mathcal{M},k,+}(1)$ such that the first $m_1$ components of the weights $w_{i,j}^{(1)}$ are non-positive for all $1 \leq i \leq m_2$, $1 \leq j \leq k$. If we set the input width $m_1 = 1$, we can encode the condition $Y_2 \succeq_2 Y_1$ as $\text{VDC}_{F_{L,\mathcal{M},k,-+}(1)}(\nu\|\mu)$, where $\nu = \mathcal{L}(Y_1)$ and $\mu = \mathcal{L}(Y_2)$ are the distributions of $Y_1, Y_2$, resp. Hence, with the appropriate Lagrange multiplier we convert problem (9) into a min-max problem between $\mu$ and the potential $u$ of VDC

$$\min_{\mu : \mu = \mathcal{L}(\langle \xi, \omega_2 \rangle)} -\int_{\mathbb{R}} x\,d\mu(x) + \lambda \text{VDC}_{F_{L,\mathcal{M},k,-+}(1)}(\nu\|\mu). \tag{10}$$

A second instance of this approach is in GAN training. Assuming that we have a baseline generator $g_0$ that can be obtained via regular training, we consider the problem:

$$\min_{g \in \mathcal{G}} \left\{ \max_{f \in \mathcal{F}} \{\mathbb{E}_{X \sim \nu_n}[f(X)] - \mathbb{E}_{Y \sim \mu_0}[f(g(Y))]\} + \lambda \text{VDC}_{F_{L,\mathcal{M},k,+}(1)}(g_\#\mu_0\|(g_0)_\#\mu_0) \right\}. \tag{11}$$

The first term in the objective function is the usual WGAN loss (Arjovsky et al., 2017), although it can be replaced by any other standard GAN loss. The second term, which is proportional to $\text{VDC}_{F_{L,\mathcal{M},k,+}(1)}(g_\#\mu_0\|(g_0)_\#\mu_0) = \max_{u \in F_{L,\mathcal{M},k,+}(1)}\{\mathbb{E}_{Y \sim \mu_0}[u(g_0(Y))] - \mathbb{E}_{Y \sim \mu_0}[u(g(Y))]\}$, enforces that $g_\#\mu_0 \succeq (g_0)_\#\mu_0$ in the Choquet order, and thus $u$ acts as a second 'Choquet' critic. Tuning $\lambda$ appropriately, the rationale is that we want a generator that optimizes the standard GAN loss, with the condition that the generated distribution dominates the baseline distribution. As stated by Proposition 1, dominance in the Choquet order translates to $g_\#\mu_0$ being more spread out than $(g_0)_\#\mu_0$, which should help avoid mode collapse and improve the diversity of generated samples. In practice, this min-max game is solved via Algorithm 1 given in App. F. For the Choquet critic, this amounts to an SGD step followed by a projection step to impose non-negativity of hidden to hidden weights.

**Generative modeling with the surrogate CT distance.** The surrogate Choquet-Toland distance is well-suited for generative modeling, as it can used in GANs in place of the usual discriminator. Namely, if $\nu$ is a target distribution, $\nu_n$ is its empirical distribution, and $\mathcal{D} = \{\mu = g_\# \mu_0 | g \in \mathcal{G}\}$ is a class of distributions that can be realized as the push-forward of a base measure $\mu_0 \in \mathcal{P}(\mathbb{R}^{d_0})$ by a function $g : \mathbb{R}^{d_0} \to \mathbb{R}^d$, the problem to solve is $g^* = \arg\min_{g \in \mathcal{G}} d_{\mathrm{CT}, F_{L,\mathcal{M},k,+}(1)}(g_\# \mu_0, \nu_n) = \arg\min_{g \in \mathcal{G}} \{\max_{u \in F_{L,\mathcal{M},k,+}(1)} \{\mathbb{E}_{X \sim \nu_n}[u(X)] - \mathbb{E}_{Y \sim \mu_0}[u(g(Y))]\} + \max_{u \in F_{L,\mathcal{M},k,+}(1)} \{\mathbb{E}_{Y \sim \mu_0}[u(g(Y))] - \mathbb{E}_{X \sim \nu_n}[u(X)]\}\}$. Algorithm 2 given in App. F summarizes learning with the surrogate CT distance.

# 7 EXPERIMENTS

**Portfolio optimization under dominance constraints** In this experiment, we use the VDC to optimize an illustrative example from Xue et al. (2020) (Example 1) that follows the paradigm laid out in Sec. 6. In this example, $\xi \in \mathbb{R} \sim P$ is drawn uniformly from $[0, 1]$, we define the benchmark portfolio as:

$$G_1(\xi) = \begin{cases} \frac{i}{20} & \xi \in [0.05 \times i, 0.05 \times (i+1)) \quad i = 0, \dots, 19 \\ 1 & \xi = 1 \end{cases}$$

and the optimization is over the parameterized portfolio $G_2(\xi) := G(\xi; z) = z\xi$.

The constrained optimization problem is thus specified as:

$$\min_z \quad -\mathbb{E}_P[G(\xi; z)] \quad \text{s.t.} \quad G(\xi; z) \succeq_2 G_1(\xi), \ 1 \le z \le 2$$

As stated in Xue et al. (2020), this example has a known solution at $z = 2$ where $\mathbb{E}_P[G_2(\xi; 2)] = 1$ outperforms the benchmark $\mathbb{E}_P[G_1(\xi)] = 0.5$. We relax the constrained optimization by including it in the objective function, thus creating min-max game (10) introduced in Sec. 6. We parameterize $F_{L,\mathcal{M},k,-+}(1)$ with a 3-layer, fully-connected, decreasing ICMN with hidden dimension 32 and maxout kernel size of 4. After 5000 steps of stochastic gradient descent on $z$ (learning rate $1\mathrm{e}^{-3}$) and the parameters of the ICMN (learning rate $1\mathrm{e}^{-3}$), using a batch size of 512 and $\lambda = 1$, we are able to attain accurate approximate values of the known solution: $z = 2$, $\frac{1}{512} \sum_{j=1}^{512} [G_z(\xi_j)] = 1.042$, and $\frac{1}{512} \sum_{j=1}^{512} [G_1(\xi_j)] = 0.496$.

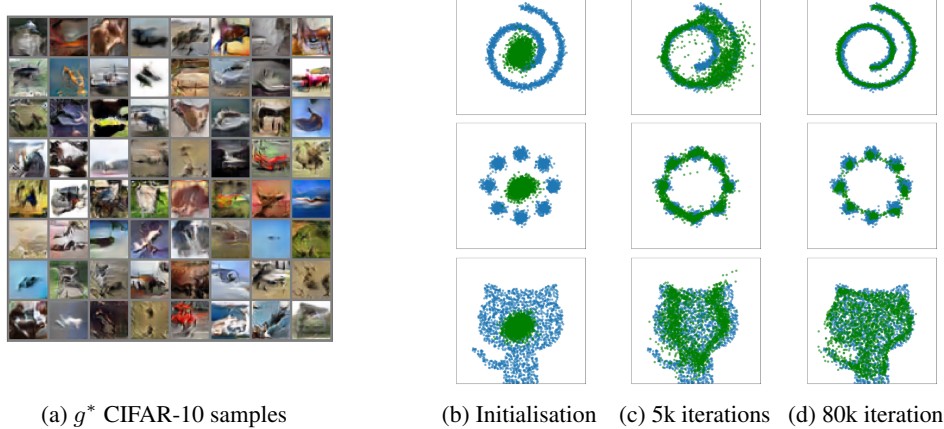

(a) $g^*$ CIFAR-10 samples     (b) Initialisation   (c) 5k iterations   (d) 80k iterations

Figure 2: Training generative models by enforcing dominance with surrogate VDC on pre-trained CIFAR-10 WGAN-GP (*Left*) and with surrogate CT distance on 2D point clouds (*Right*). Ground truth point cloud distributions (*blue*) consist of swiss roll (*Top*), circle of eight Gaussians (*Middle*), and Github icon converted to a point cloud (*Bottom*).

**Image generation with baseline model dominance** Another application of learning with the VDC is in the high-dimensional setting of CIFAR-10 (Krizhevsky & Hinton, 2009) image generation. As detailed in Sec. 6, we start by training a baseline generator $g_0$ using the regularized Wasserstein-GAN paradigm (WGAN-GP) introduced in Arjovsky et al. (2017); Gulrajani et al. (2017), where

gradient regularization is computed for the discriminator with respect to interpolates between real and generated data. When training $g^*$ and $g_0$, we used the same WGAN-GP hyperparameter configuration. We set $\lambda$ in Equation (11) to 10 (see App. F and App. H for more details). Training runs for $g^*$ and $g_0$

Table 1: FID scores for WGAN-GP and WGAN-GP with VDC surrogate for convex functions approximated by either ICNNs with softplus activations or ICMNs. ICMNs improve upon the baseline $g_0$ and outperform ICNNs with softplus. FID score for WGAN-GP + VDC includes mean values $\pm$ one standard deviation for 5 repeated runs with different random initialization seeds.

|  | FID |
| --- | --- |
| $g_0$: WGAN-GP | 69.67 |
| $g^*$: WGAN-GP + VDC CP-Flow ICNN | $83.470 \pm 3.732$ |
| $g^*$: WGAN-GP + VDC ICMN (Ours) | $\mathbf{67.317} \pm 0.776$ |

were performed in computational environments that contained 1 CPU and 1 A100 GPU. To evaluate performance of $g^*$ vs. $g_0$, we rely on the Fréchet Inception Distance (FID) introduced in Heusel et al. (2017). Note that when training a WGAN-GP baseline from scratch we used hyperparameters that potentially differ from those used in state-of-the-art implementations. Additionally, for computing FIDs we use a `pytorch-lightning` implementation of the inception network, which is different from the widely used `Tensorflow` implementation (Salimans et al., 2016), resulting in potential discrepancies in our reported baseline FID and those in the literature. FID results are reported in Table 1, where we see improved image quality from $g^*$ (as measured by lower FID) relative to the pre-trained baseline $g_0$. We therefore find that the VDC surrogate improves upon $g_0$ by providing $g^*$ with larger support, preventing mode collapse. Samples generated from $g^*$ are displayed in Figure 2a.

In order to highlight the representation power of ICMNs, we replace them in the VDC estimation by the ICNN implementation of Huang et al. (2021). Instead of maxout activation, Huang et al. (2021) uses a Softplus activation and instead of a projection step it also uses a Softplus operation to impose the non-negativity of hidden to hidden weights to enforce convexity. We see in Table 1 that VDC's estimation with ICMN outperforms Huang et al. (2021) ICNNs. While ICMNs have maximum of affine functions as a building block, ICNN's proof of universality in Huang et al. (2021) relies on approximating the latter, this could be one of the reason behind ICMN superiority.

**Probing mode collapse** To investigate how training with the surrogate VDC regularizer helps alleviate mode collapse in GAN training, we implemented GANs trained with the IPM objective alone and compared this to training with the surrogate VDC regularizer for a mixture of 8 Gaussians target distribution. In Figure 3 given in App. G we quantify mode collapse by looking at two scores: 1) the entropy of the discrete assignment of generated points to the means of the mixture 2) the negative log likelihood (NLL) of the Gaussian mixture. When training with the VDC regularizer to improve upon the collapsed generator $g_0$ (which is taken from step 55k from the unregularized GAN training), we see more stable training and better mode coverage as quantified by our scores.

**2D point cloud generation with** $d_{CT}$ We apply learning with the CT distance in a 2D generative modeling setting. Both the generator and CT critic architectures are comprised of fully-connected neural networks with maxout non-linearities of kernel size 2. Progression of the generated samples can be found in the right-hand panel of Figure 2, where we see the trained generator accurately learn the ground truth distribution. All experiments were performed in a single-CPU compute environment.

## 8 CONCLUSION

In this paper, we introduced learning with stochastic order in high dimensions via surrogate Variational Dominance Criterion and Choquet-Toland distance. These surrogates leverage input convex maxout networks, a new variant of input convex neural networks. Our surrogates have parametric statistical rates and lead to new learning paradigms by incorporating dominance constraints that improve upon a baseline. Experiments on synthetic and real image generation yield promising results. Finally, our work, although theoretical in nature, can be subject to misuse, similar to any generative method.

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

# Contents

## A  PROOFS OF SEC. 3

***Proof of Proposition 2.*** Since $\mathcal{A}$ is included in the set of convex functions, the implication from right to left is straightforward.

For the other implication, we prove the contrapositive. Suppose that $u$ is a convex function on $\Omega$ such that $\int_\Omega u \, d(\mu_- - \mu_+) > 0$. Then we show that we can construct $\tilde{u} \in \mathcal{A}$ such that $\int_\Omega \tilde{u} \, d(\mu_- - \mu_+) > 0$, which shows that the supremum over $\mathcal{A}$ is strictly positive. Denote by $\mathcal{C}$ the set of convex functions $f : \mathbb{R}^d \to \mathbb{R}$ which are the point-wise supremum of finitely many affine functions, i.e. $f(x) = \max_{i \in I}\{\langle y_i, x \rangle - a_i\}$ for some finite family $(y_i, a_i) \in \mathbb{R}^d \times \mathbb{R}$. For any convex function $g$, there is an increasing sequence $(g_n)_n \subseteq \mathcal{C}$ such that $g = \sup_n g_n$ pointwise (Ekeland & Schachermayer (2014), p. 3). Applying this to $u$, we know there exists an increasing sequence $(u_n)_n \subseteq \mathcal{C}$ such that $u = \lim_{n \to \infty} u_n$. By the dominated convergence theorem,

$$\lim_{n \to \infty} \int_\Omega u_n \, d(\mu_- - \mu_+) = \int_\Omega u \, d(\mu_- - \mu_+) > 0,$$

which means that for some $N$ large enough, $\int_\Omega u_N \, d(\mu_- - \mu_+) > 0$ as well. Since $u_N$ admits a representation $f(x) = \max_{i \in I}\{\langle y_i, x \rangle - a_i\}$ for some finite family $I$, it may be trivially extended to a convex function on $\mathbb{R}^d$.

Let $(\eta_\epsilon)_\epsilon$ be a family of non-negative radially symmetric functions in $C^2(\mathbb{R}^d)$ supported on the ball $\mathcal{B}_\epsilon(0)$ of radius $\epsilon$ centered at zero, and such that $\int_{\mathbb{R}^d} \eta_\epsilon(x) \, dx = 1$. Let $\Omega_\epsilon = \{x \in \Omega \mid \text{dist}(x, \partial\Omega) \geq \epsilon\}$. For any $x \in \mathbb{R}^d$, we have that

$$(u_N * \eta_\epsilon)(x) = \int_{\mathbb{R}^d} \eta_\epsilon(x - y) u(y) \, dy = \int_{\mathbb{R}^d} \eta_\epsilon(y') u_N(x - y') \, dy'. \tag{12}$$

By the convexity of $u_N$, we have that $u_N(\lambda x + (1 - \lambda)x' - y') \leq \lambda u_N(x - y') + (1 - \lambda)u_N(x' - y')$ for any $x, x', y' \in \mathbb{R}^d$. Thus, (12) implies that $(u_N * \eta_\epsilon)(\lambda x + (1 - \lambda)x') \leq \lambda(u_N * \eta_\epsilon)(x) + (1 - \lambda)(u_N * \eta_\epsilon)(x')$, which means that $u_N * \eta_\epsilon$ is convex. Also, by the dominated convergence theorem,

$$\lim_{\epsilon \to 0} \int_\Omega (u_N * \eta_\epsilon) \, d(\mu_- - \mu_+) = \int_\Omega u_N \, d(\mu_- - \mu_+) > 0$$

Hence, there exists $\epsilon_0 > 0$ such that $\int_\Omega (u_N * \eta_\epsilon) \, d(\mu_- - \mu_+) > 0$. Since $u_N * \eta_{\epsilon_0}$ is in $C^2(\mathbb{R}^d)$ by the properties of convolutions, its gradient is continuous. Since the closure $\bar{\Omega}$ is compact because $\Omega$ is bounded, by the Weierstrass theorem we have that

$$\sup_{x \in \overline{\Omega}} \|\nabla(u_N * \eta_{\epsilon_0})(x)\|_2 < +\infty$$

Let $r > 0$ be such that the ball $\mathcal{B}_r(0)$ is included in $\mathcal{K}$. Rescaling $u_N * \eta_{\epsilon_0}$ by an appropriate constant, we have that $\sup_{x \in \overline{\Omega}} \|\nabla(u_N * \eta_{\epsilon_0})(x)\|_2 < r$, and that means that $\nabla(u_N * \eta_{\epsilon_0})(x) \in \mathcal{K}$ for any $x \in \Omega$. Thus, $u_N * \eta_{\epsilon_0} \in \mathcal{A}$, and that means that $\sup_{u \in \mathcal{A}}\{\int_\Omega u \, d(\mu_- - \mu_+)\} > 0$, concluding the proof. $\square$

**Lemma 1.** *Let $\Omega = [-1,1]^d$ and $\mathcal{K} = \{x \in \mathbb{R}^d \mid \|x\|_2 \leq C\}$. The function class $\mathcal{A} = \{u : \Omega \to \mathbb{R}, u \text{ convex and } \nabla u \text{ a.e. } \in \mathcal{K}\}$ is equal to the space $F_d(M,C)$ of convex functions on $[-1,1]^d$ such that $|f(x)| \leq M$ and $|f(x) - f(y)| \leq C\|x - y\|$ for any $x, y \in [-1,1]^d$, up to a constant term. Here, $M = C\sqrt{d}$.*

*Proof.* Looking at problem (1), note that adding a constant to a function $u \in \mathcal{A}$ does not change the value of the objective function. Thus, we can add the restriction that $u(0) = 0$, and since $\Omega$ is compact and has Lipschitz constant upper-bounded by $\sup_{x \in K} \|x\|$, any such function $u$ must fulfill $M := \sup_{u \in \mathcal{A}} \sup_{x \in \Omega} |u(x)| < +\infty$. Thus, we have that functions in $\mathcal{A}$ belong to $\{u : \Omega \to \mathbb{R}, u \text{ convex and } \nabla u \text{ a.e. } \in K, \sup_{x \in \Omega} |u(x)| \leq M\}$ up to a constant term, for a well chosen $M$.

Now we will use the particular form of $\Omega$ and $\mathcal{K}$. First, note that we can take $M = C\sqrt{d}$ without loss of generality. Given $u \in \mathcal{A}$, we have that $\|\nabla u(x)\|_2 \leq C$ for a.e. $x \in [-1,1]^d$. By the mean value theorem, we have that $|u(x) - u(y)| = |\int_0^1 \langle \nabla f(tx + (1-t)y), x - y \rangle \, dt| \leq C\|x - y\|$, implying that $u$ is $C$-Lipschitz. This shows that $A \subseteq F_d(M,C)$. Rademacher's theorem states that $C$-Lipschitz functions are a.e. differentiable, and gradient norms must be upper-bounded by $C$ wherever gradients exist as otherwise one reaches a contradiction. Hence, $F_d(M,C) \subseteq A$, concluding the proof. □

**Lemma 2** (Metric entropy of convex functions, Bronshtein (1976), Thm. 6). *Let $F_d(M,C)$ be the compact space of convex functions on $[-1,1]^d$ such that $|f(x)| \leq M$ and $|f(x) - f(y)| \leq C\|x - y\|$ for any $x, y \in [-1,1]^d$. The metric entropy of this space with respect to the uniform norm topology satisfies*

$$\log N(\delta; F_d(M,C), \|\cdot\|_\infty) \leq K\delta^{-\frac{d}{2}},$$

*for some constant $\mathcal{K}$ that depend on $C$, $M$ and $d$.*

**Lemma 3** (Dudley's entropy integral bound, Wainwright (2019), Thm. 5.22, Dudley (1967)). *Let $\{X_\theta \mid \theta \in \mathbb{T}\}$ be a zero-mean sub-Gaussian process with respect the metric $\rho_X$ on $\mathbb{T}$. Let $D = \sup_{\theta, \theta' \in \mathbb{T}} \rho_X(\theta, \theta')$. Then for any $\delta \in [0,D]$ such that $N(\delta; \mathbb{T}, \rho_X) \geq 10$, we have*

$$\mathbb{E}[\sup_{\theta, \theta' \in \mathbb{T}}(X_\theta - X_{\theta'})] \leq \mathbb{E}\left[\sup_{\substack{\gamma, \gamma' \in \mathbb{T} \\ \rho_X(\gamma, \gamma') \leq \delta}} (X_\gamma - X_{\gamma'})\right] + 32 \int_\delta^D \sqrt{\log N(t; \mathbb{T}, \rho_X)} \, dt.$$

**Proposition 4.** *For any family of $n$ points $(X_i)_{i=1}^n \subseteq [-1,1]^d$, the empirical Rademacher complexity of the function class $F_d(M,C)$ satisfies*

$$\mathbb{E}_\epsilon[\|\mathbb{S}_n\|_{F_d(M,C)}] \leq Kn^{-\frac{2}{d}},$$

*where $\mathcal{K}$ is a constant depending on $M$, $C$ and $d$.*

*Proof.* We choose $\mathbb{T}_n = \{(f(X_i))_{i=1}^n \in \mathbb{R}^n \mid f \in F_d(M,C)\}$, we define the Rademacher process $X_f = \sum_{i=1}^n \epsilon_i f(X_i)$, which is sub-Gaussian with respect to the metric $\rho_n(f, f') = \sqrt{\sum_{i=1}^n (f(X_i) - f'(X_i))^2}$. Remark that $D \leq 2M\sqrt{n}$. For any $\delta \in [0,D]$, we apply Lemma 3 setting $f' \equiv 0$ and we get

$$\mathbb{E}_\epsilon[\|\mathbb{S}_n\|_{F_d(M,C)}] = \frac{1}{n}\mathbb{E}\left[\sup_{f \in \mathbb{T}_n} X_{f'}\right]$$

$$\leq \frac{1}{n}\left(\mathbb{E}\left[\sup_{\substack{f, f' \in \mathbb{T}_n \\ \rho_n(f, f') \leq \delta}} (X_f - X_{f'})\right] + 32 \int_\delta^D \sqrt{\log N(t; F_d(M,C), \rho_n)} \, dt\right). \quad (13)$$

Note that for any $f, f' \in F_d(M,C)$, $\rho_n(f, f') \leq \sqrt{n}\|f - f'\|_\infty$, which means that $\log N(\delta; F_d(M,C), \rho_n) \leq \log N(\delta/\sqrt{n}; F_d(M,C), \|\cdot\|_\infty)$. Thus, Lemma 2 implies that

$$\log N(\delta; F_d(M,C), \rho_n) \leq K\left(\frac{\delta}{\sqrt{n}}\right)^{-\frac{d}{2}} = K\delta^{-\frac{d}{2}}n^{\frac{d}{4}},$$

Hence,

$$\int_\delta^D \sqrt{\log N(t; F_d(M,C), \rho_n)} \, dt \leq \int_\delta^D \sqrt{K}n^{\frac{d}{8}}t^{-\frac{d}{4}} \, dt = \left[\frac{\sqrt{K}n^{\frac{d}{8}}}{-\frac{d}{4}+1}t^{-\frac{d}{4}+1}\right]_\delta^D \quad (14)$$

$$\leq \frac{\sqrt{K}n^{\frac{d}{8}}}{\frac{d}{4}+1}(\delta^{-\frac{d}{4}+1} - (2M\sqrt{n})^{-\frac{d}{4}+1})$$

We set $\delta = n^{\frac{1}{2} - \frac{2}{d}}$, and we get that $\delta^{-\frac{d}{4}+1} = n^{(\frac{1}{2} - \frac{2}{d})(-\frac{d}{4}+1)} = n^{-\frac{d}{8}+1-\frac{2}{d}}$. Hence, the right-hand side of (14) is upper-bounded by $\frac{\sqrt{K}}{\frac{d}{4}+1} n^{1-\frac{2}{d}}$. And since $\sum_{i=1}^{n} \epsilon_i(f(X_i) - f'(X_i)) \leq \sum_{i=1}^{n} |f(X_i) - f'(X_i)| \leq \sqrt{n}\rho_n(f, f')$, we have

$$\mathbb{E}\left[\sup_{\substack{\gamma,\gamma' \in \mathbb{T} \\ \rho_X(\gamma,\gamma') \leq \delta}} (X_\gamma - X_{\gamma'})\right] \leq \sqrt{n}\delta = \sqrt{n}n^{\frac{1}{2}-\frac{2}{d}} = n^{1-\frac{2}{d}}.$$

Plugging these bounds back into (13), we obtain

$$\mathbb{E}_\epsilon[\|\mathbb{S}_n\|_{F_d(M,C)}] \leq \left(\frac{\sqrt{K}}{\frac{d}{4}+1} + 1\right)n^{-\frac{2}{d}}.$$

Since $\mathcal{K}$ already depends on $d$, we rename it as $\mathcal{K} \leftarrow \frac{\sqrt{K}}{\frac{d}{4}+1} + 1$, concluding the proof. $\qquad\square$

***Proof of Theorem 2.*** Let $F_d(C)$ be the space of convex functions on $[-1, 1]^d$ such that $|f(x)| \leq C\sqrt{d}$ and $|f(x) - f(y)| \leq C\|x - y\|$ for any $x, y \in [-1, 1]^d$. Lemma 1 shows that when $\Omega = [-1, 1]^d$ and $\mathcal{K} = \{x \in \mathbb{R}^d \mid \|x\|_2 \leq C\}$, functions in $\mathcal{A}$ belong to $F_d(C)$ up to a constant term, which means that $\text{VDC}_{\mathcal{A}}(\mu\|\mu) = \sup_{u \in F_d(C)} \int_\Omega u\, d(\mu_- - \mu_+)$. Theorem 11 of Sriperumbudur et al. (2009) shows that for any function class $\mathcal{F}$ on $\Omega$ such that $r := \sup_{f \in \mathcal{F}, x \in \Omega} |f(x)| < +\infty$, with probability at least $1 - \delta$ we have

$$|\sup_{f \in \mathcal{F}}\{\mathbb{E}_\mu f(x) - \mathbb{E}_\mu f(x)\} - \sup_{f \in \mathcal{F}}\{\mathbb{E}_{\mu_n} f(x) - \mathbb{E}_{\mu_n} f(x)\}|$$

$$\leq \sqrt{18r^2 \log(\tfrac{\delta}{4})(\tfrac{1}{\sqrt{m}} + \tfrac{1}{\sqrt{n}})} + 2\hat{\mathcal{R}}_n(\mathcal{F}, (x_i)_{i=1}^n) + 2\hat{\mathcal{R}}_n(\mathcal{F}, (y_i)_{i=1}^n),$$

where $\hat{\mathcal{R}}_n$ denotes the empirical Rademacher complexity. Proposition 4 shows that $\hat{\mathcal{R}}_n(F_d(C), (x_i)_{i=1}^n) \leq \mathcal{K}n^{-\frac{2}{d}}$ for any $(x_i)_{i=1}^n \in \Omega \subseteq [-1, 1]^d$, where $\mathcal{K}$ depends on $C$ and $d$. This concludes the proof. $\qquad\square$

## B  PROOFS OF SEC. 4

**Proposition 5.** *Input convex maxout networks (Definition 3) are convex with respect to their input.*

*Proof.* The proof is by finite induction. We show that for any $2 \leq \ell \leq L-1$ and $1 \leq i \leq m_\ell$, the function $x \mapsto x_i^{(\ell)}$ is convex. The base case $\ell = 2$ holds because $x \mapsto x_i^{(2)} = \frac{1}{\sqrt{m_2}} \max_{j \in [k]} \langle w_{i,j}^{(1)}, (x, 1) \rangle$ is a pointwise supremum of convex (affine) functions, which is convex. For the induction case, we have that $x \mapsto x_i^{(\ell-1)}$ is convex for any $1 \leq i \leq m_{\ell-1}$ by the induction hypothesis. Since a linear combination of convex functions with non-negative coefficients is convex, we have that for any $1 \leq i \leq m_\ell$, $1 \leq j \leq k$, $x \mapsto \langle w_{i,j}^{(\ell-1)}, (x^{(\ell-1)}, 1) \rangle$ is convex. Finally, $x \mapsto x_i^{(\ell)} = \frac{1}{\sqrt{m_\ell}} \max_{j \in [k]} \langle w_{i,j}^{(\ell-1)}, (x^{(\ell-1)}, 1) \rangle$ is convex because it is the pointwise supremum of convex functions. $\qquad\square$

***Proof of Proposition 3.*** We can reexpress $f(x)$ as:

$$f(x) = \frac{1}{\sqrt{m_L}} \sum_{i=1}^{m_L} a_i \langle w_{i,j_{L-1,i}^*}^{(L-1)}, (x^{(L-1)}, 1) \rangle, \quad j_{L-1,i}^* = \arg\max_{j \in [k]} \langle w_{i,j}^{(L-1)}, (x^{(L-1)}, 1) \rangle \quad (15)$$

$$x_i^{(\ell)} = \frac{1}{\sqrt{m_\ell}} \langle w_{i,j_{\ell-1,i}^*}^{(\ell-1)}, (x^{(\ell-1)}, 1) \rangle, \quad j_{\ell-1,i}^* = \arg\max_{j \in [k]} \langle w_{i,j_{\ell-1,i}^*}^{(\ell-1)}, (x^{(\ell-1)}, 1) \rangle.$$

For $1 \leq \ell \leq L-1$, we define the matrices $W_\ell^* \in \mathbb{R}^{m_{\ell+1}, m_\ell}$ such that their $i$-th row is the vector $[\frac{1}{\sqrt{m_{\ell+1}}} w_{i,j_{\ell,i}^*}^{(\ell)}]_{1:m_\ell}$, i.e. the vector containing the first $m_\ell$ components of $\frac{1}{\sqrt{m_{\ell+1}}} w_{i,j_{\ell,i}^*}^{(\ell)}$. Iterating the chain rule, one can see that for almost every $x \in \mathbb{R}^d$, [1]

$$\nabla f(x) = (W_1^*)^\top (W_2^*)^\top \ldots (W_{L-1}^*)^\top a$$

---

[1] The gradient of $f$ is well defined when there exists a neighborhood of $x$ for which $f$ is an affine function.

Since the spectral norm $\| \cdot \|_2$ is sub-multiplicative and $\|A\|_2 = \|A^\top\|_2$, we have that $\|\nabla f(x)\| = \|(W_1^*)^\top (W_2^*)^\top \cdots (W_{L-1}^*)^\top\|_2 \|a\| = \|W_1^*\|_2 \|W_2^*\|_2 \cdots \|W_{L-1}^*\|_2 \|a\|$. We compute the Frobenius norm of $W_\ell^*$:

$$\|W_\ell^*\|_F^2 = \tfrac{1}{m_\ell} \sum_{i=1}^{m_\ell} \left\| [w_{i,j_{\ell,i}^*}^{(\ell)}]_{1:m_\ell} \right\|^2 \leq \tfrac{1}{m_\ell} \sum_{i=1}^{m_\ell} \left\| w_{i,j_{\ell,i}^*}^{(\ell)} \right\|^2 \leq 1.$$

Since for any matrix $A$, $\|A\|_2 \leq \|A\|_F$, and the vector $a$ satisfies $\|a\| \leq 1$, we obtain that $\|\nabla f(x)\| \leq 1$. To obtain the bound on $|f(x)|$, we use again the expression (15). For $1 \leq \ell \leq L-1$, we let $b_\ell \in \mathbb{R}^{m_{\ell+1}}$ be the vector such that the $i$-th component is $[\frac{1}{\sqrt{m_{\ell+1}}} w_{i,j_{\ell,i}^*}^{(\ell)}]_{m_{\ell+1}}$. Since $|[\frac{1}{\sqrt{m_{\ell+1}}} w_{i,j_{\ell,i}^*}^{(\ell)}]_{m_{\ell+1}}| \leq \frac{1}{\sqrt{m_{\ell+1}}}$, $\|b_\ell\| \leq 1$. It is easy to see that

$$f(x) = a^\top W_{L-1}^* \cdots W_1^* x + \sum_{\ell=1}^{L-1} a^\top W_{L-1}^* \cdots W_{\ell+1}^* b_\ell.$$

Thus,

$$|f(x)| \leq \|a\| \left( \|W_{L-1}^* \cdots W_1^* x\| + \sum_{\ell=1}^{L-1} \|W_{L-1}^* \cdots W_{\ell+1}^* b_\ell\| \right) \leq L.$$

The bound on $\|x^{(\ell)}\|$ follows similarly, as $x^\ell = W_{\ell-1}^* \cdots W_1^* x + \sum_{\ell'=1}^{\ell-1} a^\top W_{\ell-1}^* \cdots W_{\ell+1}^* b_\ell$. $\qquad \square$

**Proposition 6.** *Let $F_{L,\mathcal{M},k}(1)$ be the subset of $F_{L,\mathcal{M},k}$ such that for all $1 \leq \ell \leq L-1$, $1 \leq i \leq m_\ell$, $1 \leq j \leq k$, $\|w_{i,j}^{(\ell)}\|_2 \leq 1$, and $\|a\|_2 = \sum_{i=1}^{m_L} a_i^2 \leq 1$. For any $f \in F_{L,\mathcal{M},k}(1)$ and $x \in \mathcal{B}_1(\mathbb{R}^d)$, we have that $|f(x)| \leq 1$. The metric entropy of $F_{L,\mathcal{M},k}(1)$ with respect to $\tilde{\rho}_n$ admits the upper bound:*

$$\log N\left(\delta; F_{L,\mathcal{M},k}(1), \tilde{\rho}_n\right) \leq \sum_{\ell=2}^{L} k m_\ell (m_{\ell-1} + 1) \log\left(1 + \tfrac{2^{3+L-\ell}}{\delta}\right) + m_L \log\left(1 + \tfrac{4}{\delta}\right)$$

*Proof.* We define the function class $G_{L,\mathcal{M},k}$ that contains the functions from $\mathbb{R}^d$ to $\mathbb{R}^{m_L}$ of the form

$$g(x) = \left( \tfrac{1}{\sqrt{m_L}} \max_{j \in [k]} \langle w_{i,j}^{(L-1)}, (x^{(L-1)}, 1) \rangle \right)_{i=1}^{m_L},$$

$$\forall 2 \leq \ell \leq L-1, \quad x_i^{(\ell)} = \tfrac{1}{\sqrt{m_\ell}} \max_{j \in [k]} \langle w_{i,j}^{(\ell-1)}, (x^{(\ell-1)}, 1) \rangle, \quad x^{(1)} = x,$$

$$\forall 1 \leq \ell \leq L-1, 1 \leq i \leq m_\ell, 1 \leq j \leq k, \quad \|w_{i,j}^{(\ell)}\|_2 \leq 1.$$

Given $\{X_i\}_{i=1}^n \subseteq \mathcal{B}_1(\mathbb{R}^d)$ we define the pseudo-metric $\tilde{\rho}_n$ between functions from $\mathbb{R}^d$ to $\mathbb{R}^{m_L}$ as $\tilde{\rho}_n(f, f') = \sqrt{\tfrac{1}{n} \sum_{i=1}^{m_L} \sum_{j=1}^{n} (f_i(X_j) - f_i'(X_j))^2}$.

We prove by induction that

$$\log N(\delta; G_{L,\mathcal{M},k}, \tilde{\rho}_n) \leq \sum_{\ell=2}^{L} k m_\ell (m_{\ell-1} + 1) \log\left(1 + \tfrac{2^{2+L-\ell}\sqrt{(\ell-1)^2+1}}{\delta}\right)$$

To show the induction case, note that for $L \geq 3$, any $g \in G_{L,\mathcal{M},k}$ can be written as

$$g(x) = \left( \tfrac{1}{\sqrt{m_L}} \max_{j \in [k]} \langle w_{i,j}^{(L-1)}, (h(x), 1) \rangle \right)_{i=1}^{m_L},$$

where $h \in G_{L-1,\mathcal{M},k}$. Remark that given a $\frac{\delta}{2\sqrt{(L-1)^2+1}}$-cover $\mathcal{C}'$ of $\mathcal{B}_1(\mathbb{R}^{m_{L-1}+1})$, there exist $\tilde{w}_{i,j}^{(L-1)} \in \mathcal{C}'$ such that $\|\tilde{w}_{i,j}^{(L-1)} - w_{i,j}^{(L-1)}\| \leq \frac{\delta}{2\sqrt{(L-1)^2+1}}$. Hence, if $\tilde{h}$ is such that $\tilde{\rho}_n(\tilde{h}, h) \leq \frac{\delta}{2}$,

and we define $\tilde{g}(x) = (\frac{1}{\sqrt{2m_L}} \max_{j\in[k]} \langle \tilde{w}_{i,j}^{(L-1)}, (\tilde{h}(x), 1)\rangle)_{i=1}^{m_L}$, we obtain

$\tilde{\rho}_n(\tilde{g}, g)^2$

$= \frac{1}{nm_L} \sum_{i=1}^{m_L} \sum_{k=1}^{n} \left( \max_{j\in[k]} \langle \tilde{w}_{i,j}^{(L-1)}, (\tilde{h}(X_k), 1)\rangle - \max_{j\in[k]} \langle w_{i,j}^{(L-1)}, (h(X_k), 1)\rangle \right)^2$

$\leq \frac{1}{nm_L} \sum_{i=1}^{m_L} \sum_{k=1}^{n} \max_{j\in[k]} \left( \langle \tilde{w}_{i,j}^{(L-1)} - w_{i,j}^{(L-1)}, (\tilde{h}(X_k), 1)\rangle - \langle w_{i,j}^{(L-1)}, (h(X_k) - \tilde{h}(X_k), 0)\rangle \right)^2$

$\leq \frac{2}{nm_L} \sum_{i=1}^{m_L} \sum_{k=1}^{n} \max_{j\in[k]} \left( \langle \tilde{w}_{i,j}^{(L-1)} - w_{i,j}^{(L-1)}, (\tilde{h}(X_k), 1)\rangle \right)^2 + \left( \langle w_{i,j}^{(L-1)}, (h(X_k) - \tilde{h}(X_k), 0)\rangle \right)^2$

$\leq \frac{2}{nm_L} \sum_{i=1}^{m_L} \sum_{k=1}^{n} \max_{j\in[k]} \|\tilde{w}_{i,j}^{(L-1)} - w_{i,j}^{(L-1)}\|^2 \|(\tilde{h}(X_k), 1)\|^2 + \|w_{i,j}^{(L-1)}\|^2 \|h(X_k) - \tilde{h}(X_k)\|^2$

$\leq \frac{2}{nm_L} \sum_{i=1}^{m_L} \sum_{k=1}^{n} \max_{j\in[k]} \|\tilde{w}_{i,j}^{(L-1)} - w_{i,j}^{(L-1)}\|^2 \|(\tilde{h}(X_k), 1)\|^2 + \|w_{i,j}^{(L-1)}\|^2 \|h(X_k) - \tilde{h}(X_k)\|^2$

$\leq \frac{1}{nm_L} \sum_{i=1}^{m_L} \sum_{k=1}^{n} \left( \frac{\delta^2}{4(L^2+1)} (L^2 + 1) + \|h(X_k) - \tilde{h}(X_k)\|^2 \right)$

$\leq \frac{1}{nm_L} \sum_{i=1}^{m_L} \sum_{k=1}^{n} \frac{\delta^2}{2} + \frac{1}{n} \sum_{k=1}^{n} \|h(X_k) - \tilde{h}(X_k)\|^2 \leq \delta^2.$

In the second-to-last inequality we used that if $h \in G_{\ell, \mathcal{M}, k}$, $\|h(x)\| \leq \ell$. This is equivalent to the bound $\|x^{(\ell)}\| \leq \ell$ shown in Proposition 3. Hence, we can build a $\delta$-cover of $G_{L, \mathcal{M}, k}$ in the pseudo-metric $\tilde{\rho}_n$ from the Cartesian product of a $\frac{\delta}{2}$-cover of $G_{L-1, \mathcal{M}, k}$ in $\tilde{\rho}_n$ and $km_L$ copies of a $\frac{\delta}{2\sqrt{(L-1)^2+1}}$-cover of $\mathcal{B}_1(\mathbb{R}^{m_{L-1}+1})$ in the $\|\cdot\|_2$ norm. Thus,

$$N(\delta; G_{L, \mathcal{M}, k}, \tilde{\rho}_n) \leq N\left(\frac{\delta}{2}; G_{L-1, \mathcal{M}, k}, \tilde{\rho}_n\right) \cdot N\left(\frac{\delta}{2\sqrt{(L-1)^2+1}}; \mathcal{B}_1(\mathbb{R}^{m_{L-1}+1}), \|\cdot\|_2\right)^{km_L}.$$

The metric entropy of the unit ball admits the upper bound $\log N(\delta; \mathcal{B}_1(\mathbb{R}^d), \|\cdot\|_2) \leq d \log(1 + \frac{2}{\delta})$ (Wainwright (2019), Example 5.8). Consequently,

$\log N(\delta; G_{L, \mathcal{M}, k}, \tilde{\rho}_n)$

$\leq \log N\left(\frac{\delta}{2}; G_{L-1, \mathcal{M}, k}, \tilde{\rho}_n\right) + km_L \log N\left(\frac{\delta}{2\sqrt{(L-1)^2+1}}; \mathcal{B}_1(\mathbb{R}^{m_{L-1}+1}), \|\cdot\|_2\right)$

$\leq \sum_{\ell=2}^{L-1} km_\ell(m_{\ell-1} + 1) \log\left(1 + \frac{2^{2+L-\ell}\sqrt{(\ell-1)^2+1}}{\delta}\right) + km_L(m_{L-1} + 1) \log\left(1 + \frac{4\sqrt{L^2+1}}{\delta}\right)$

$= \sum_{\ell=2}^{L} km_\ell(m_{\ell-1} + 1) \log\left(1 + \frac{2^{2+L-\ell}\sqrt{(\ell-1)^2+1}}{\delta}\right)$

In the second inequality we used the induction hypothesis.

To conclude the proof, note that an arbitrary function $f \in F_{L, \mathcal{M}, k}(1)$ can be written as $f(x) = \langle a, g(x)\rangle$, where $a \in \mathcal{B}_1(\mathbb{R}^{m_L})$ and $g \in G_{L, \mathcal{M}, k}$. Applying an analogous argument, we see that a $\frac{\delta}{2\sqrt{L^2+1}}$-cover of $\mathcal{B}_1(\mathbb{R}^{m_L})$ and a $\frac{\delta}{2}$-cover of $G_{L, \mathcal{M}, k}$ give rise to a $\delta$-cover of $F_{L, \mathcal{M}, k}(1)$. Hence,

$$\log N\left(\delta; F_{L, \mathcal{M}, k}(1), \tilde{\rho}_n\right) \leq \log N\left(\frac{\delta}{2}; G_{L, \mathcal{M}, k}, \tilde{\rho}_n\right) + \log N\left(\frac{\delta}{2}; \mathcal{B}_1(\mathbb{R}^{m_L}), \|\cdot\|_2\right)$$

$$\leq \sum_{\ell=2}^{L} km_\ell(m_{\ell-1} + 1) \log\left(1 + \frac{2^{3+L-\ell}\sqrt{(\ell-1)^2+1}}{\delta}\right) + m_L \log\left(1 + \frac{4\sqrt{L^2+1}}{\delta}\right).$$

Finally, to show the bound $|f(x)| \leq 1$ for all $f \in F_{L, \mathcal{M}, k}(1)$ and $x \in \mathcal{B}_1(\mathbb{R}^d)$, we use that $|f(x)| \leq |\langle a, g(x)\rangle| \leq \|a\|\|g(x)\| \leq 1$ and that if $g \in G_{\ell, \mathcal{M}, k}$ and $x \in \mathcal{B}_1(\mathbb{R}^d)$, then $\|g(x)\| \leq 1$, as shown before. □

**Proposition 7.** *Suppose that for all $1 \leq \ell \leq L$, the widths $m_\ell$ satisfy $m_\ell \leq m$. Then, the Rademacher complexity of the class $F_{L,\mathcal{M},k}(1)$ satisfies:*

$$\mathbb{E}_\epsilon[\|\mathbb{S}_n\|_{F_{L,\mathcal{M},k}(1)}] \leq 64\sqrt{\tfrac{(L-1)km(m+1)}{n}}\left(\sqrt{(L+1)\log(2) + \tfrac{1}{2}\log(L^2+1)} + \tfrac{\sqrt{\pi}}{2}\right). \quad (16)$$

*Proof.* We apply Dudley's entropy integral bound (Lemma 3). We choose $\mathbb{T}_n = \{(f(X_i))_{i=1}^n \in \mathbb{R}^n \mid f \in F_{L,\mathcal{M},k}(1)\}$, we define the Rademacher process $X_f = \frac{1}{\sqrt{n}}\sum_{i=1}^n \epsilon_i f(X_i)$, which is sub-Gaussian with respect to the metric $\tilde{\rho}_n(f, f') = \sqrt{\frac{1}{n}\sum_{i=1}^n (f(X_i) - f'(X_i))^2}$. Remark that $D \leq 2$. Setting $f' \equiv 0$ and $\delta = 0$ in Lemma 3, we obtain that

$$\mathbb{E}_\epsilon[\|\mathbb{S}_n\|_{F_{L,\mathcal{M},k}(1)}] = \tfrac{1}{\sqrt{n}}\mathbb{E}\left[\sup_{f \in \mathbb{T}_n} X_{f'}\right]$$
$$\leq \tfrac{32}{\sqrt{n}}\int_0^2 \sqrt{\log N(t; F_{L,\mathcal{M},k}(1), \tilde{\rho}_n)}\, dt.$$

Applying the metric entropy bound from Proposition 6, we get

$$\sqrt{\log N(\delta; F_{L,\mathcal{M},k}(1), \tilde{\rho}_n)}$$
$$\leq \sqrt{\sum_{\ell=2}^L km_\ell(m_{\ell-1}+1)\log\left(1 + \tfrac{2^{3+L-\ell}\sqrt{(\ell-1)^2+1}}{\delta}\right) + m_L \log\left(1 + \tfrac{4\sqrt{L^2+1}}{\delta}\right)}$$
$$\leq \sqrt{(L-1)km(m+1)\log\left(1 + \tfrac{2^{L+1}\sqrt{L^2+1}}{\delta}\right)}$$
$$\leq \sqrt{(L-1)km(m+1)\log\left(\tfrac{2^{L+2}\sqrt{L^2+1}}{\delta}\right)}$$

In the last equality we used that for $\delta \in [0, 2]$, $1 + \tfrac{2^{L+1}\sqrt{L^2+1}}{\delta} \leq \tfrac{2^{L+2}\sqrt{L^2+1}}{\delta}$. We compute the integral

$$\int_0^2 \sqrt{\log\left(\tfrac{2^{L+2}\sqrt{L^2+1}}{t}\right)}\, dt = 2^{L+2}\sqrt{L^2+1}\int_0^{2^{-(L+1)}\frac{1}{\sqrt{L^2+1}}} \sqrt{-\log(t')}\, dt'.$$

Applying Lemma 4 with $z = 2^{-(L+1)}\frac{1}{\sqrt{L^2+1}}\frac{1}{\sqrt{L^2+1}}$, we obtain that

$$\int_0^{2^{-(L+1)}\frac{1}{\sqrt{L^2+1}}} \sqrt{-\log(t')}\, dt' \leq 2^{-(L+1)}\left(\sqrt{(L+1)\log(2) + \tfrac{1}{2}\log(L^2+1)} + \tfrac{\sqrt{\pi}}{2}\right).$$

Putting everything together yields equation (16). $\qquad\square$

**Lemma 4.** *For any $z \in (0, 1]$, we have that*

$$\int_0^z \sqrt{-\log(x)}\, dx = z\sqrt{-\log(z)} + \tfrac{\sqrt{\pi}}{2}\mathrm{erfc}(\sqrt{-\log(z)}) \leq z\left(\sqrt{-\log(z)} + \tfrac{\sqrt{\pi}}{2}\right).$$

*Here, $\mathrm{erfc}$ denotes the complementary error function, defined as $\mathrm{erfc}(x) = \frac{2}{\sqrt{\pi}}\int_x^{+\infty} e^{-t^2}\, dt$.*

*Proof.* We rewrite the integral as:

$$\int_0^z \int_0^{-\log(x)} \tfrac{1}{2\sqrt{y}}\, dy\, dx = \int_0^{-\log(z)} \tfrac{1}{2\sqrt{y}}\int_0^z dx\, dy + \int_{-\log(z)}^{+\infty} \tfrac{1}{2\sqrt{y}}\int_0^{e^{-y}} dx\, dy$$
$$= z\int_0^{-\log(z)} \tfrac{1}{2\sqrt{y}}\, dy + \int_{-\log(z)}^{+\infty} \tfrac{e^{-y}}{2\sqrt{y}}\, dy$$
$$= z\sqrt{-\log(z)} + \int_{\sqrt{-\log(z)}}^{+\infty} e^{-t^2}\, dt$$
$$= z\sqrt{-\log(z)} + \tfrac{\sqrt{\pi}}{2}\mathrm{erfc}(\sqrt{-\log(z)})$$

The complementary error function satisfies the bound $\mathrm{erfc}(x) \leq e^{-x^2}$ for any $x > 0$, which implies the final inequality. $\qquad\square$

**Proof of Theorem 3.** Proposition 7 proves that under the condition $m_\ell \leq m$, the empirical Rademacher complexity of the class $F_{L,\mathcal{M},k}(1)$ satisfies $\hat{R}_n(F_{L,\mathcal{M},k}(1)) \leq 64\sqrt{\frac{(L-1)km(m+1)}{n}}(\sqrt{(L+1)\log(2) + \frac{1}{2}\log(L^2+1)} + \frac{\sqrt{\pi}}{2})$. Applying Theorem 11 of Sriperumbudur et al. (2009) as in the proof of Theorem 2, and that for any $f \in F_{L,\mathcal{M},k}(1)$ and $x \in \mathcal{B}_r(\mathbb{R}^d)$, we have that $|f(x)| \leq r$ (see Proposition 6), we obtain the result. $\square$

## C  PROOFS OF SEC. 5

**Proof of Theorem 4.** To show (i), we have that for any $\nu' \in \mathcal{P}(\mathcal{K})$,

$$0 = \frac{1}{2}W_2^2(\mu_+, \nu') - \frac{1}{2}W_2^2(\mu_-, \nu') + \frac{1}{2}W_2^2(\mu_-, \nu') - \frac{1}{2}W_2^2(\mu_+, \nu')$$

$$\leq \sup_{\nu \in \mathcal{P}(\mathcal{K})}\left\{\frac{1}{2}W_2^2(\mu_+, \nu) - \frac{1}{2}W_2^2(\mu_-, \nu)\right\} + \sup_{\nu \in \mathcal{P}(\mathcal{K})}\left\{\frac{1}{2}W_2^2(\mu_-, \nu) - \frac{1}{2}W_2^2(\mu_+, \nu)\right\} = d_{\text{CT},\mathcal{A}}(\mu_+, \mu_-).$$

The right-to-left implication of (ii) is straight-forward. To show the left-to-right one, we use the definition for the CT distance, rewriting $\text{VDC}_\mathcal{A}(\mu_+ \| \mu_-)$ and $\text{VDC}_\mathcal{A}(\mu_- \| \mu_+)$ in terms of their definitions:

$$d_{\text{CT},\mathcal{A}}(\mu_+, \mu_-) = \sup_{u \in \mathcal{A}}\left\{\int_\Omega u \, d(\mu_+ - \mu_-)\right\} + \sup_{u \in \mathcal{A}}\left\{\int_\Omega u \, d(\mu_+ - \mu_-)\right\}. \tag{17}$$

Since the two terms in the right-hand side are non-negative, $d_{\text{CT},\mathcal{A}}(\mu_+, \mu_-) = 0$ implies that they are both zero. Then, applying Proposition 2, we obtain that $\mu_- \preceq \mu_+$ and $\mu_+ \preceq \mu_-$ according to the Choquet order. The antisymmetry property of partial orders then implies that $\mu_+ = \mu_-$. To show (iii), we use equation (17) again. The result follows from

$$\sup_{u \in \mathcal{A}}\left\{\int_\Omega u \, d(\mu_1 - \mu_2)\right\} \leq \sup_{u \in \mathcal{A}}\left\{\int_\Omega u \, d(\mu_1 - \mu_3)\right\} + \sup_{u \in \mathcal{A}}\left\{\int_\Omega u \, d(\mu_3 - \mu_2)\right\},$$

$$\sup_{u \in \mathcal{A}}\left\{\int_\Omega u \, d(\mu_2 - \mu_1)\right\} \leq \sup_{u \in \mathcal{A}}\left\{\int_\Omega u \, d(\mu_2 - \mu_3)\right\} + \sup_{u \in \mathcal{A}}\left\{\int_\Omega u \, d(\mu_3 - \mu_1)\right\}.$$

$\square$

## D  PROOFS OF SUBSEC. 5.1

**Lemma 5.** *For a function class $\mathcal{F}$ that contains the zero function, define $\|\mathbb{S}_n\|_\mathcal{F} = \sup_{f \in \mathcal{F}}|\frac{1}{n}\sum_{i=1}^n \epsilon_i f(X_i)|$, where $\epsilon_i$ are Rademacher variables. $\mathbb{E}_{X,\epsilon}[\|\mathbb{S}_n\|_\mathcal{F}]$ is known as the Rademacher complexity. Suppose that $0$ belongs to the compact set $\mathcal{K}$. We have that*

$$\frac{1}{2}\mathbb{E}_{X,\epsilon}[\|\mathbb{S}_n\|_{\bar{\mathcal{A}}}] \leq \mathbb{E}[d_{CT,\mathcal{A}}(\mu, \mu_n)] \leq 4\mathbb{E}_{X,\epsilon}[\|\mathbb{S}_n\|_\mathcal{A}],$$

*where $\bar{\mathcal{A}} = \{f - \mathbb{E}_\mu[f] \mid f \in \mathcal{A}\}$ is the centered version of the class $\mathcal{A}$.*

*Proof.* We will use an argument similar to the proof of Prop. 4.11 of Wainwright (2019) (with the appropriate modifications) to obtain the Rademacher complexity upper and lower bounds. We start

with the lower bound:

$$\mathbb{E}_{X,\epsilon}[\|\mathbb{S}_n\|_{\bar{\mathcal{A}}}] = \mathbb{E}_{X,\epsilon}\left[\sup_{f\in\mathcal{A}}\left|\frac{1}{n}\sum_{i=1}^{n}\epsilon_i(f(X_i) - \mathbb{E}_{Y_i}[f(Y_i)])\right|\right]$$

$$\leq \mathbb{E}_{X,Y,\epsilon}\left[\sup_{f\in\mathcal{A}}\left|\frac{1}{n}\sum_{i=1}^{n}\epsilon_i(f(X_i) - f(Y_i))\right|\right] = \mathbb{E}_{X,Y}\left[\sup_{f\in\mathcal{A}}\left|\frac{1}{n}\sum_{i=1}^{n}(f(X_i) - f(Y_i))\right|\right]$$

$$\leq \mathbb{E}_X\left[\sup_{f\in\mathcal{A}}\left|\frac{1}{n}\sum_{i=1}^{n}(f(X_i) - \mathbb{E}[f])\right|\right] + \mathbb{E}_Y\left[\sup_{f\in\mathcal{A}}\left|\frac{1}{n}\sum_{i=1}^{n}(f(Y_i) - \mathbb{E}[f])\right|\right]$$

$$\leq 2\mathbb{E}_X\left[\sup_{f\in\mathcal{A}}\frac{1}{n}\sum_{i=1}^{n}(f(X_i) - \mathbb{E}[f])\right] + 2\mathbb{E}_X\left[\sup_{f\in\mathcal{A}}\frac{1}{n}\sum_{i=1}^{n}(\mathbb{E}[f] - f(X_i))\right]$$

$$= 2\mathbb{E}[\text{VDC}_{\mathcal{A}}(\mu_n\|\mu) + \text{VDC}_{\mathcal{A}}(\mu\|\mu_n)].$$

The last inequality follows from the fact that $\sup_{f\in\mathcal{A}}|\frac{1}{n}\sum_{i=1}^{n}(f(X_i) - \mathbb{E}[f])| \leq \sup_{f\in\mathcal{A}}\frac{1}{n}\sum_{i=1}^{n}(f(X_i) - \mathbb{E}[f]) + \sup_{f\in\mathcal{A}}\frac{1}{n}\sum_{i=1}^{n}(\mathbb{E}[f] - f(X_i))$, which holds as long as the two terms in the right-hand side are non-negative. This happens when $f \equiv 0$ belongs to $\mathcal{A}$ as a consequence of $0 \in \mathcal{K}$.

The upper bound follows essentially from the classical symmetrization argument:

$$\mathbb{E}[d_{\text{CT},\mathcal{A}}(\mu,\mu_n)] \leq 2\mathbb{E}_X\left[\sup_{f\in\mathcal{A}}\left|\frac{1}{n}\sum_{i=1}^{n}(f(X_i) - \mathbb{E}[f])\right|\right] \leq 4\mathbb{E}_{X,\epsilon}\left[\sup_{f\in\mathcal{A}}\left|\frac{1}{n}\sum_{i=1}^{n}\epsilon_i f(X_i)\right|\right].$$

$\square$

**Lemma 6** (Relation between Rademacher and Gaussian complexities, Exercise 5.5, Wainwright (2019)). *Let $\|\mathbb{G}_n\|_{\mathcal{F}} = \sup_{f\in\mathcal{F}}|\frac{1}{n}\sum_{i=1}^{n}z_i f(X_i)|$, where $z_i$ are standard Gaussian variables. $\mathbb{E}_{X,z}[\|\mathbb{G}_n\|_{\mathcal{F}}]$ is known as the Gaussian complexity. We have*

$$\frac{\mathbb{E}_{X,z}[\|\mathbb{G}_n\|_{\mathcal{F}}]}{2\sqrt{\log n}} \leq \mathbb{E}_{X,\epsilon}[\|\mathbb{S}_n\|_{\mathcal{F}}] \leq \sqrt{\frac{\pi}{2}}\mathbb{E}_{X,z}[\|\mathbb{G}_n\|_{\mathcal{F}}].$$

Given a set $\mathbb{T} \subseteq \mathbb{R}^n$, the family of random variables $\{G_\theta, \theta \in \mathbb{T}\}$, where $G_\theta := \langle \omega, \theta \rangle = \sum_{i=1}^{n}\omega_i\theta_i$ and $\omega_i \sim N(0,1)$ i.i.d., defines a stochastic process is known as the canonical Gaussian process associated with $\mathbb{T}$.

### D.1 RESULTS USED IN THE LOWER BOUND OF THEOREM 5

**Lemma 7** (Sudakov minoration, Wainwright (2019), Thm. 5.30; Sudakov (1973)). *Let $\{G_\theta, \theta \in \mathbb{T}\}$ be a zero-mean Gaussian process defined on the non-empty set $\mathbb{T}$. Then,*

$$\mathbb{E}\left[\sup_{\theta\in\mathbb{T}} G_\theta\right] \geq \sup_{\delta>0}\frac{\delta}{2}\sqrt{\log M_G(\delta;\mathbb{T})}.$$

*where $M_G(\delta;\mathbb{T})$ is the $\delta$-packing number of $\mathbb{T}$ in the metric $\rho_G(\theta,\theta') = \sqrt{\mathbb{E}[(X_\theta - X_{\theta'})^2]}$.*

**Proposition 8.** *Let $C_0, C_1$ be universal constants independent of the dimension $d$, and suppose that $n \geq C_1\log(d)$. Recall that $F_d(M,C)$ is the set of convex functions on $[-1,1]^d$ such that $|f(x)| \leq M$ and $|f(x) - f(y)| \leq C\|x - y\|$ for any $x, y \in [-1,1]^d$, and that $\bar{F}_d(M,C) = \{f - \mathbb{E}_\mu[f]\,|\,f \in F_d(M,C)\}$. The Gaussian complexity of the set $\overline{F_d(M,C)}$ satisfies*

$$\mathbb{E}_{X,z}[\|\mathbb{G}_n\|_{\overline{F_d(M,C)}}] \geq C_0\sqrt{\frac{C}{d(1+C)}}n^{-\frac{2}{d}}.$$

*Proof.* Given $(X_i)_{i=1}^n \subseteq \mathbb{R}^d$, let $\mathbb{T}_n = \{(f(X_i))_{i=1}^n \in \mathbb{R}^n \mid f \in \overline{F_d(M,C)}\}$. Let $X_i$ be sampled i.i.d. from the uniform measure on $[-1,1]^d$. We have that with probability at least $1/2$ on the instantiation of $(X_i)_{i=1}^n$,

$$\mathbb{E}_z[\|\mathbb{G}_n\|_{\overline{F_d(M,C)}}] = \tfrac{1}{n}\mathbb{E}\left[\sup_{\theta \in \mathbb{T}_n} X_\theta\right]$$

$$\geq \frac{1}{2n}\sqrt{\frac{Cn}{128d(1+C)}}n^{-\frac{2}{d}}\left(\log M\left(\sqrt{\frac{Cn}{128d(1+C)}}n^{-\frac{2}{d}}; \overline{F_d(M,C)}, \rho_n\right)\right)^{1/2}$$

$$\gtrsim \frac{1}{2n}\sqrt{\frac{Cn}{128d(1+C)}}n^{-\frac{2}{d}}\left(\log\left(\frac{2^{\frac{n}{16}}}{32n^{\frac{2}{d}}\sqrt{2d(1+C)Cn+1}}\right)\right)^{1/2}$$

$$\approx \frac{1}{2n}\sqrt{\frac{Cn}{128d(1+C)}}n^{-\frac{2}{d}}\left(\frac{n}{16}\log(2) - \log(32\sqrt{2d(1+C)C}) - \left(\tfrac{1}{2}+\tfrac{2}{d}\right)\log(n)\right)^{1/2}$$

$$\geq C_0\sqrt{\frac{C}{d(1+C)}}n^{-\frac{2}{d}}.$$

The first inequality follows from Sudakov minoration (Lemma 7) by setting $\delta = \sqrt{\frac{Cn}{128d(1+C)}}n^{-\frac{2}{d}}$.
The second inequality follows from the lower bound on the packing number of $\bar{\mathcal{A}}$ given by Corollary 1. In the following approximation we neglected the term 1 in the numerator inside of the logarithm. In the last inequality, $C_0$ is a universal constant independent of the dimension $d$. The inequality holds as long as $\log(32\sqrt{2d(1+C)C}) + (\tfrac{1}{2}+\tfrac{2}{d})\log(n) \leq \frac{n}{32}\log(2)$, which is true when $n \geq C_1\log(d)$ for some universal constant $C_1$. $\qquad\square$

**Proposition 9.** *With probability at least (around) $1/2$ on the instantiation of $(X_i)_{i=1}^n$ as i.i.d. samples from the uniform distribution on $\mathbb{S}^{d-1}$, the packing number of the set $F_d(M,C)$ with respect to the metric $\rho_n(f,f') = \sqrt{\sum_{i=1}^n (f(X_i) - f'(X_i))^2}$ satisfies*

$$M\left(\sqrt{\frac{Cn}{32d(1+C)}}n^{-\frac{2}{d}}; F_d(M,C), \rho_n\right) \gtrsim 2^{n/16}.$$

*Proof.* This proof uses the same construction of Thm. 6 of Bronshtein (1976), which shows a lower bound on the metric entropy of $F_d(M,C)$ in the $L^\infty$ norm. His result follows from associating a subset of convex functions in $F_d(M,C)$ to a subset of convex polyhedrons, and then lower-bounding the metric entropy of this second set. Note that the pseudo-metric $\rho_n$ is weaker than the $L^\infty$ norm, which means that our result does not follow from his. Instead, we need to use a more intricate construction for the set of convex polyhedrons, and rely on the Varshamov-Gilbert lemma (Lemma 8).

First, we show that if we sample $n$ points $(X_i)_{i=1}^n$ i.i.d. from the uniform distribution over the unit ball of $\mathbb{R}^d$, with constant probability (around $\frac{1}{2}$) there exists a $\delta$-packing of the unit ball of $\mathbb{R}^d$ with $k \approx \frac{\delta^{-d/2}}{2}$ points. For any $n \in \mathbb{Z}^+$, we define the set-valued random variable

$$S_n = \{i \in \{1,\dots,n\} \mid \forall 1 \leq j < i, \|X_i - X_j\|_2 \geq \delta\},$$

and the random variable $A_n = |S_n|$. That is, $A_n$ is the number of points $X_i$ that are at least epsilon away of any point with a lower index; clearly the set of such points constitutes a $\delta$-packing of the unit ball of $\mathbb{R}^d$. We have that $\mathbb{E}[A_n|(X_i)_{i=1}^{n-1}] = A_{n-1} + \Pr(\forall 1 \leq i \leq n-1, \|X_n - X_i\|_2 \geq \delta)$. Since for a fixed $X_i$ and uniformly distributed $X_n$, $\Pr(\|X_n - X_i\|_2 \leq \delta) \leq \delta^d$, a union bound shows that $\Pr(\forall 1 \leq i \leq n-1, \|X_n - X_i\|_2 \geq \delta) \geq 1 - (k-1)\delta^d$. Thus, by the tower property of conditional expectations:

$$\mathbb{E}[A_n] \geq \mathbb{E}[A_{n-1}] + 1 - (n-1)\delta^d.$$

A simple induction shows that $\mathbb{E}[A_n] \geq n - \sum_{i=0}^{n-1} i\delta^d = n - \frac{n(n-1)\delta^d}{2}$. This is a quadratic function of $n$. For a given $\delta > 0$, we choose $n$ that maximizes this expression, which results in

$$1 - \frac{(2n-1)\delta^d}{2} \sim 0 \implies n \sim \delta^{-d} + \tfrac{1}{2} \tag{18}$$

$$\implies \mathbb{E}[A_n] \gtrsim \delta^{-d} + \tfrac{1}{2} + \frac{(\delta^{-d}+\frac{1}{2})(\delta^{-d}-\frac{1}{2})\delta^d}{2} = \delta^{-d} + \tfrac{1}{2} + \frac{\delta^{-d}-\frac{1}{4}\delta^d}{2} = \frac{\delta^{-d}}{2} + \tfrac{1}{2} - \frac{\delta^d}{8}.$$

where we used $\sim$ instead of $=$ to remark that $n$ must be an integer. Since $A_n$ is lower-bounded by 1 and upper-bounded by $n \sim \delta^{-d} + \frac{1}{2}$, Markov's inequality shows that

$$P(A_n \geq \mathbb{E}[A_n]) \gtrsim \frac{\frac{\delta^{-d}}{2} + \frac{1}{2} - \frac{\delta^d}{8} - 1}{\delta^{-d} - \frac{1}{2}} \approx \frac{1}{2},$$

where the approximation works for $\delta \ll 1$. Taking $k = A_n$, this shows the existence of a $\delta$-packing of the unit ball of size $\approx \frac{\delta^{-d/2}}{2}$ with probability at least (around) $\frac{1}{2}$.

Next, we use a construction similar to the proof of the lower bound in Thm. 6 of Bronshtein (1976). That is, we consider the set $\tilde{S}_n = \{(x, g(x)) \mid x \in S_n\} \subseteq \mathbb{R}^{d+1}$, where $g : [-1, 1]^d \to \mathbb{R}$ is the map that parameterizes part of the surface of the $(d-1)$-dimensional hypersphere $\mathbb{S}^{d-1}(R, t)$ centered at $(0, \ldots, 0, t)$ of radius $R$ for well chosen $t, R$.

As in the proof of Thm. 6 of Bronshtein (1976), if $x \in \tilde{S}_n \subseteq \mathbb{S}^{d-1}(R, t)$ and we let $S^{d-1}(R, t)_x$ denote the tangent space at $x$, we construct a hyperplane $P$ that is parallel to $\mathbb{S}^{d-1}(R, t)_x$ at a distance $\epsilon < R$ from the latter and that intersects the sphere. A simple trigonometry exercise involving the lengths of the chord and the sagitta of a two-dimensional circular segment shows that any point in $\mathbb{S}^{d-1}(R, t)$ that is at least $\sqrt{2R\epsilon}$ away from $x$ will be separated from $x$ by the hyperplane $P$. Thus, the convex hull of $\tilde{S}_n \setminus \{x\}$ is at least $\epsilon$ away from $x$.

For any subset $S' \subseteq \tilde{S}_n$, we define the function $g_{S'}$ as the function whose epigraph (the set of points on or above the graph of a convex function) is equal to the convex hull of $S'$. Note that $g_{S'}$ is convex and piecewise affine. Let us set $\delta = \sqrt{2R\epsilon}$. If $x$ is a point in $\tilde{S}_n \setminus S'$, by the argument in the previous paragraph the convex hull of $S'$ is at least $\epsilon$ away from $x$. Thus, $|g_{S'}(x) - g_{S' \cup \{x\}}| \geq \epsilon$. The functions $g_{S'}$ and $g_{S''}$ will differ by at least $\epsilon$ at each point in the symmetric difference $S' \Delta S''$.

By the Varshamov-Gilbert lemma (Lemma 8), there is a set of $2^{k/8}$ different subsets $S'$ such that $g_{S'}$ differ pairwise at at least $k/4$ points in $\tilde{S}_n$ by at least $\epsilon$. Thus, for any $S' \neq S''$ in this set, we have that

$$\rho_n(g_{S'}, g_{S''}) = \sqrt{\sum_{i=1}^n (g_{S'}(X_i) - g_{S'}(X_i))^2} \geq \sqrt{\frac{k}{4}} \epsilon.$$

We have to make sure that all the functions $g_{S'}$ belong to the set $F_d(M, C)$. Since $|\frac{d}{dx} \sqrt{R^2 - x^2}| = \frac{|x|}{\sqrt{R^2 - x^2}}$, the function $g$ will be $C$-Lipschitz on $[-1, 1]^d$ if we take $R^2 \geq \frac{d(1+C)}{C}$, and in that case $g_{S'}$ will be Lipschitz as well for any $S' \subseteq \tilde{S}_n$. To make sure that the uniform bound $\|g_{S'}\|_\infty \leq M$ holds, we adjust the parameter $t$.

To obtain the statement of the proposition we start from a certain $n$ and set $\delta$ such that (18) holds: $\delta = n^{-\frac{1}{d}}$, and we set $k \approx \frac{n}{2}$. Since $\delta = \sqrt{2R\epsilon}$, we have that $\epsilon = \frac{1}{2R} \delta^2 \approx \frac{1}{2} \sqrt{\frac{C}{d(1+C)}} n^{-\frac{2}{d}}$, which means that

$$2^{k/8} \approx 2^{n/16}, \qquad \sqrt{\frac{k}{4}} \epsilon \approx \sqrt{\frac{Cn}{32d(1+C)}} n^{-\frac{2}{d}}.$$

$\square$

**Lemma 8** (Varshamov-Gilbert). *Let $N \geq 8$. There exists a subset $\Omega \subseteq \{0, 1\}^N$ such that $|\Omega| \geq 2^{N/8}$ and for any $x, x' \in \Omega$ such that $x \neq x'$, at least $N/4$ of the components differ.*

**Lemma 9.** *For any $\epsilon > 0$, the packing number of the centered function class $\overline{F_d(M, C)} = \{f - \mathbb{E}_\mu[f] \mid f \in F_d(M, C)\}$ with respect to the metric $\rho_n$ fulfills*

$$M(\epsilon/2; \overline{F_d(M, C)}, \rho_n) \geq M(\epsilon; F_d(M, C), \rho_n)/(4nC/\epsilon + 1),$$

*Proof.* Let $\mathcal{S}$ be an $\epsilon$-packing of $F_d(M, C)$ with respect to $\rho_n$ (i.e. $|\mathcal{S}| = M(\epsilon; F_d(M, C), \rho_n)$). Let $F_d(M, C, t, \delta) = \{f \in F_d(M, C) \mid |\mathbb{E}_\mu[f] - t| \leq \delta\}$. If we let $(t_i)_{i=1}^m$ be a $\delta$ packing of $[-C, C]$, we can write $F_d(M, C) = \cup_{i=1}^m F_d(M, C, t_i, \delta)$.

By the pigeonhole principle, there exists an index $i \in \{1, \ldots, m\}$ such that $|\mathcal{S} \cap F_d(M, C, t_i, \delta)| \geq |S|/m$. Let $F_d(M, C, t_i) = \{f \in F_d(M, C) \,|\, \mathbb{E}_\mu[f] = t_i\}$, and let $P : F_d(M, C) \to F_d(M, C, t_i)$ be the projection operator defined as $f \mapsto f - E_\mu[f] + t_i$.

If we take $\delta = \epsilon/(4n)$, we have that $\rho_n(Pf, Pf') \geq \epsilon/2$ for any $f \neq f' \in \mathcal{S} \cap F_d(M, C, t_i, \delta)$, as $\rho_n(Pf, Pf') \leq \rho_n(Pf, f) + \rho_n(f, f') + \rho_n(f', Pf')$, and $\rho_n(f, Pf) \leq n|E_\mu[f] - t_i| \leq n\delta = \epsilon/4$, while $\rho_n(f, f') \geq \epsilon$. Thus, the $\epsilon/2$-packing number of $F_d(M, C, t_i)$ is lower-bounded by $|\mathcal{S} \cap F_d(M, C, t_i, \delta)| \geq |S|/m$. Since $m = \lfloor (2C + 2\delta)/(2\delta) \rfloor \leq 4nC/\epsilon + 1$, this is lower-bounded by $|S|/(4nC/\epsilon + 1)$.

The proof concludes using the observation that the map $F_d(M, C, t_i) \to F_d(\bar{M}, C)$ defined as $f \mapsto f - t_i$ is an isometric bijection with respect to $\rho_n$. $\qquad\square$

**Corollary 1.** *With probability at least $1/2$, the packing number of the set $\overline{F_d(M, C)}$ with respect to the metric $\rho_n$ satisfies*

$$M\left(\sqrt{\frac{Cn}{128d(1+C)}}n^{-\frac{2}{d}}; \overline{F_d(M, C)}, \rho_n\right) \geq 2^{\frac{n}{16}} \frac{1}{32n^{\frac{2}{d}}\sqrt{2d(1+C)Cn+1}}$$

***Proof of Theorem 5.*** Lemma 5 provides upper and lower bounds to $\mathbb{E}[d_{\mathrm{CT}, \mathcal{A}}(\mu, \mu_n)]$ in terms of the Rademacher complexities of $\mathcal{A}$ and its centered version, $\bar{\mathcal{A}}$. Lemma 1 in App. A states that $\mathcal{A}$ is equal to the space $F_d(M, C)$ of convex functions on $[-1, 1]^d$ such that $|f(x)| \leq M$ and $|f(x) - f(y)| \leq C\|x - y\|$ for any $x, y \in [-1, 1]^d$. Let $\mathbb{E}_{X,\epsilon}[\|\mathbb{S}_n\|_{\mathcal{F}}]$, $\mathbb{E}_{X,\epsilon}[\|\mathbb{G}_n\|_{\mathcal{F}}]$ be the Rademacher and Gaussian complexities of a function class $\mathcal{F}$ (see definitions in Lemma 5 and Lemma 6). We can write

$$\mathbb{E}_{X,\epsilon}[\|\mathbb{S}_n\|_{\bar{\mathcal{A}}}] \geq \frac{\mathbb{E}_{X,z}[\|\mathbb{G}_n\|_{\bar{\mathcal{A}}}]}{2\sqrt{\log n}} \geq C_0\sqrt{\frac{C}{2d(1+C)\log(n)}}n^{-\frac{2}{d}},$$

where the first inequality holds by Lemma 6, and the second inequality follows from Proposition 8. This gives rise to (7) upon redefining $C_0 \leftarrow C_0/\sqrt{8}$. Equation (8) follows from the Rademacher complexity upper bound in Lemma 5, and the upper bound on the empirical Rademacher complexity of $F_d(M, C)$ given by Proposition 4. $\qquad\square$

***Proof of Theorem 6.*** We upper-bound $\mathbb{E}[d_{\mathrm{CT}, F_{L,\mathcal{M},k,+}(1)}(\mu, \mu_n)]$ as in the upper bound of $\mathbb{E}[d_{\mathrm{CT}, \mathcal{A}}(\mu, \mu_n)]$ in Lemma 5:

$$\mathbb{E}[d_{\mathrm{CT}, F_{L,\mathcal{M},k,+}(1)}(\mu, \mu_n)] \leq 2\mathbb{E}_X\left[\sup_{f \in F_{L,\mathcal{M},k,+}(1)} \left|\tfrac{1}{n}\sum_{i=1}^n (f(X_i) - \mathbb{E}[f])\right|\right]$$

$$\leq 4\mathbb{E}_{X,\epsilon}\left[\sup_{f \in F_{L,\mathcal{M},k,+}(1)} \left|\tfrac{1}{n}\sum_{i=1}^n \epsilon_i f(X_i)\right|\right] = 4\mathbb{E}_\epsilon[\|\mathbb{S}_n\|_{F_{L,\mathcal{M},k,+}(1)}].$$

Likewise, we get that $\mathbb{E}[\mathrm{VDC}_{F_{L,\mathcal{M},k,+}(1)}(\mu, \mu_n)] \leq 2\mathbb{E}_\epsilon[\|\mathbb{S}_n\|_{F_{L,\mathcal{M},k,+}(1)}]$. Since $F_{L,\mathcal{M},k,+}(1) \subset F_{L,\mathcal{M},k}(1)$, we have that $\mathbb{E}_\epsilon[\|\mathbb{S}_n\|_{F_{L,\mathcal{M},k,+}(1)}] \leq \mathbb{E}_\epsilon[\|\mathbb{S}_n\|_{F_{L,\mathcal{M},k}(1)}]$. Then, the result follows from the Rademacher complexity upper bound from Proposition 7. $\qquad\square$

# E  SIMPLE EXAMPLES IN DIMENSION 1 FOR VDC AND $d_{CT}$

For simple distributions over compact sets of $\mathbb{R}$, we can compute the CT discrepancy exactly. Let $\eta : \mathbb{R} \to \mathbb{R}$ be a non-negative bump-like function, supported on $[-1, 1]$, symmetric w.r.t 0, increasing on $(-1, 0]$, and such that $\int_{\mathbb{R}} \eta = 1$. Clearly $\eta$ is the density of some probability measure with respect to the Lebesgue measure.

**Same variance, different mean.** Let $\mu_+ \in \mathcal{P}(\mathbb{R})$ with density $\eta(x - a)$ for some $a > 0$, and let $\mu_- \in \mathcal{P}(\mathbb{R})$ with density $\eta(x + a)$. We let $\mathcal{K} = [-C, C]$ for any $C > 0$.

**Proposition 10.** *Let $F(x) = \int_{-\infty}^{x}(\eta(y+a) - \eta(y-a))\,dy$, and $G(x) = \int_0^x F(y)\,dy$. We obtain that $\mathrm{VDC}_K(\mu_+\|\mu_-) = 2CG(+\infty)$. The optimum of $(\mathcal{P}_2)$ is equal to the Dirac delta at $-C$: $\nu = \delta_{-C}$, while an optimum of $(\mathcal{P}_1)$ is $u = -Cx$. We have that $\mathrm{VDC}_K(\mu_+\|\mu_-) = 2CG(+\infty)$ as well, and thus, $d_{CT,\mathcal{A}}(\mu_-, \mu_+) = 4CG(+\infty)$.*

*Proof.* Suppose that first that $a > 0$. We have that $F(-\infty) = F(-a - 1) = 0, F(+\infty) = F(a + 1) = 0$. $F$ is even, and $\sup_x |F(x)| = 1$. We have that $G$ is odd (in particular $G(-a - 1) = -G(a + 1)$) and non-decreasing on $R$. Also, $0 < G(+\infty) = G(a + 1) \leq a + 1$. Let $\mathcal{K} = [-C, C]$ where $C > 0$, and let $\Omega = [-a - 1, a + 1]$. For any twice-differentiable[2] convex function $u$ on $\Omega$ such that $u' \in \mathcal{K}$, we have

$$\int_\Omega u(x)\,d(\mu_- - \mu_+)(y) = \int_\Omega u(x)(\eta(y + a) - \eta(y - a))\,dy = [u(x)F(x)]_{-a-1}^{a+1} - \int_\Omega u'(x)F(x)\,dy$$
$$= -\int_\Omega u'(x)F(x)\,dy = [-u'(x)G(x)]_{-a-1}^{a+1} + \int_\Omega u''(x)G(x)\,dy$$
$$= -G(a + 1)(u'(a + 1) + u'(-a - 1)) + \int_\Omega u''(x)G(x)\,dy$$

If we reexpress $u(a + 1) = u'(-a - 1) + \int_{-a-1}^{a+1} u''(x)\,dx$, the right-hand side becomes

$$-2G(a + 1)u'(-a - 1) + \int_\Omega u''(x)(G(x) - G(a + 1))\,dy \tag{19}$$

Since $G$ is increasing, for any $x \in [-a - 1, a + 1), G(x) < G(a + 1)$. The convexity assumption on $u$ implies that $u'' \geq 0$ on $\Omega$, and the condition $u' \in \mathcal{K}$ means that $u' \in [-C, C]$. Thus, the function $u$ that maximizes (19) fulfills $u' \equiv -C, u'' \equiv 0$. Thus, we can take $u = -Cx$. The measure $\nu = (\nabla u)_{\#}\mu_- = (\nabla u)_{\#}\mu_+$ is equal to the Dirac delta at $-C$: $\nu = \delta_{-C}$. Hence,

$$\sup_{u \in \mathcal{A}} \int_\Omega u\,d(\mu_- - \mu_+) = 2CG(a + 1).$$

Thus, $\mathrm{VDC}_K(\mu_+\|\mu_-) = 2CG(a + 1)$. Reproducing the same argument yields $\mathrm{VDC}_K(\mu_-\|\mu_+) = 2CG(a + 1)$, and hence the CT distance is equal to $d_{CT,\mathcal{A}}(\mu_-, \mu_+) = 4CG(a + 1)$. $\square$

**Same mean, different variance.** Let $\mu_+ \in \mathcal{P}(\mathbb{R})$ with density $\frac{1}{a}\eta(\frac{x}{a})$ for some $a > 0$, and let $\mu_- \in \mathcal{P}(\mathbb{R})$ with density $\eta(x)$. Note that $\mu_+$ has support $[-a, a]$, and standard deviation equal to $a$ times the standard deviation of $\mu_-$. We let $\mathcal{K} = [-C, C]$ for any $C > 0$ as before.

**Proposition 11.** *Let $F(x) = \int_{-\infty}^{x}(\eta(y) - \frac{1}{a}\eta(\frac{y}{a}))\,dy$, and $G(x) = \int_{-\infty}^{x} F(y)\,dy$. When $a < 1$, we have that $\mathrm{VDC}_K(\mu_+\|\mu_-) = 2CG(0)$. The optimum of $(\mathcal{P}_2)$ is equal to $\nu = \frac{1}{2}\delta_{-C} + \frac{1}{2}\delta_C$, while an optimum of $(\mathcal{P}_1)$ is $u = C|x|$. When $a > 1$, $\mathrm{VDC}_K(\mu_+\|\mu_-) = 0$. For any $a > 0$, $d_{CT,\mathcal{A}}(\mu_-, \mu_+) = 2CG(0)$.*

*Proof.* We define $\Omega = [-\max\{a, 1\}, \max\{a, 1\}]$ and $\mathcal{K} = [-C, C]$. We have that $F(-\infty) = F(-\max\{a, 1\}) = 0, F(+\infty) = F(\max\{a, 1\}) = 0$, and $F(0) = 0$. $F$ is odd. When $a < 1$ it is non-positive on $[0, +\infty)$ and non-negative on $(-\infty, 0]$. When $a > 1$, it is non-negative on $[0, +\infty)$ and non-positive on $(-\infty, 0]$. We have that $G(-\infty) = G(-\max\{a, 1\}) = 0$ and $G(+\infty) = G(\max\{a, 1\}) = 0$. $G$ is even. When $a < 1$, $G$ is non-negative, non-decreasing on $(-\infty, 0]$ and non-increasing on $[0, +\infty)$: it has a global maximum at $0$. When $a > 1$, $G$ is non-positive, non-increasing on $(-\infty, 0]$ and non-decreasing on $[0, +\infty)$: it has a global minimum at $0$. We have that

$$\int_\Omega u(x)\,d(\mu_- - \mu_+)y = \int_\Omega u(x)\left(\eta(y) - \frac{1}{a}\eta\left(\frac{y}{a}\right)\right)dy = [u(x)F(x)]_{-a-1}^{a+1} - \int_\Omega u'(x)F(x)\,dy$$
$$= -\int_\Omega u'(x)F(x)\,dy = [-u'(x)G(x)]_{-a-1}^{a+1} + \int_\Omega u''(x)G(x)\,dy = \int_\Omega u''(x)G(x)\,dy. \tag{20}$$

Thus, when $a < 1$ this expression is maximized when $u'' \propto \delta_0$. Taking into account the constraints $u' \in [-C, C]$, the optimal $u'$ is $u'(x) = C\mathrm{sign}(x)$, which means that an optimal $u$ is $u(x) = C|x|$.

---

[2]Using a mollifier sequence, any convex function can be approximated arbitrarily well by a twice-differentiable convex function.

Thus, the measure $\nu = (\nabla u)_{\#}\mu_- = (\nabla u)_{\#}\mu_+$ is equal to the average of Dirac deltas at $-C$ and $C$: $\nu = \frac{1}{2}\delta_{-C} + \frac{1}{2}\delta_C$. We obtain that

$$\sup_{u \in \mathcal{A}} \int_\Omega u \, d(\mu_- - \mu_+) = 2CG(0).$$

When $a > 1$, the expression (20) is maximized when $u'' = 0$, which means that any $u'$ constant and any $u$ affine work. Any measure $\nu$ concentrated at a point in $[-C, C]$ is optimal. Thus, $\sup_{u \in \mathcal{A}} \int_\Omega u \, d(\mu_- - \mu_+) = 0$. We conclude that $d_{\mathrm{CT},\mathcal{A}}(\mu_+, \mu_-) = 2CG(0)$. $\qquad\square$

## F  ALGORITHMS FOR LEARNING DISTRIBUTIONS WITH SURROGATE VDC AND CT DISTANCE

In Algorithms 1 and 2, we present the steps for learning with the VDC and the CT distance, respectively. In order to enforce convexity of the ICMNs, we leverage the projected gradient descent algorithm after updating hidden neural network parameters. Additionally, to regularize the $u$ networks in Algorithm 1, we include a term that penalizes the square of the outputs of the Choquet critic on the baseline and generated samples (Mroueh & Sercu, 2017).

---

**Algorithm 1** Enforcing dominance constraints with the surrogate VDC

---

**Input:** Target distribution $\nu$, baseline $g_0$, latent distribution $\mu_0$, integer maxEpochs, integer discriminatorEpochs, Choquet weight $\lambda$, GAN learning rate $\eta$, Choquet critic learning rate $\eta_{\mathrm{VDC}}$, Choquet critic regularization weight $\lambda_{u,reg}$, WGAN gradient penalty regularization weight $\lambda_{\mathrm{GP}}$
**Initialize:** untrained discriminator $f_\varphi$, untrained generator $g_\theta$, untrained Choquet ICMN critic $u_\psi$
$\psi \leftarrow \mathrm{ProjectHiddenWeightsToNonNegative}()$
**for** $i = 1$ **to** maxEpochs **do**
  $L_{\mathrm{WGAN}}(\varphi, \theta) = \mathbb{E}_{Y \sim \mu_0}[f_\varphi(g_\theta(Y))] - \mathbb{E}_{X \sim \nu}[f_\varphi(X)]$
  $L_{\mathrm{GP}} = \mathbb{E}_{t \sim \mathrm{Unif}[0,1]}[\mathbb{E}_{X \sim \nu, Y \sim \mu_0}(||\nabla_x f_\varphi(tg_\theta(Y) + (1-t)X)|| - 1)^2]$
  $L_{\mathrm{VDC}}(\psi, \theta) = \mathbb{E}_{Y \sim \mu_0}[u_\psi(g_\theta(Y)) - u_\psi(g_0(Y))]$
  **for** $j = 1$ **to** discriminatorEpochs **do**
    $\varphi \leftarrow \varphi - \eta\nabla_\varphi(L_{\mathrm{WGAN}} + \lambda_{\mathrm{GP}}L_{\mathrm{GP}})$ {ADAM optimizer}
    $L_{\mathrm{VDC}_{reg}} = \mathbb{E}_{Y \sim \mu_0}[u_\psi(g_\theta(Y))^2 + u_\psi(g_0(Y))^2]$
    $\psi \leftarrow \psi - \eta_{\mathrm{VDC}}\nabla_\psi(L_{\mathrm{VDC}} + \lambda_{u,reg}L_{\mathrm{VDC}_{reg}})$ {ADAM optimizer}
    $\psi \leftarrow \mathrm{ProjectHiddenWeightsToNonNegative}()$
  **end for**
  $\theta \leftarrow \theta + \eta\nabla_\theta(L_{\mathrm{WGAN}} + \lambda L_{\mathrm{VDC}})$ {ADAM optimizer}
**end for**
Return $g_\theta$

---

## G  PROBING MODE COLLAPSE

As described in Sec. 7, to investigate how training with the surrogate VDC regularizer helps alleviate mode collapse in GAN training, we implemented GANs trained with the IPM objective alone and compared this to training with the surrogate VDC regularizer for a mixture of 8 Gaussians target distribution. To track mode collapse, we report two metrics: 1) The mode collapse score that we define as follows: for each generated point, we assign it to the nearest neighbor cluster in the target mixture and obtain a histogram over the modes computed on all generated points. The closer this histogram is to the uniform distribution, the less mode collapsed is the generator. We quantify this with the KL distance of this histogram to the uniform distribution on 8 modes. 2) The negative log likelihood (NLL) of the Gaussian mixture. A converged generator needs to have a low negative likelihood and low mode collapse score. In Figure 3, we see that in the baseline training (unregularized GAN training), we observe mode collapse and cycling between modes, evidenced by the fluctuating mode collapse score and NLL. In contrast, when training with the VDC regularizer to improve upon the collapse generator $g_0$ (which is taken from step 55k from the unregularized GAN training), we see more stable training and better mode coverage. As the regularization weight $\lambda_{\mathrm{VDC}}$ for VDC increases, the dominance constraint is more strongly enforced resulting in a better NLL and smaller mode collapse score.

---

**Algorithm 2** Generative modeling with the surrogate CT distance

---

**Input:** Target distribution $\nu$, latent distribution $\mu_0$, integer maxEpochs, integer criticEpochs, learning rate $\eta$
**Initialize:** untrained generator $g_\theta$, untrained Choquet ICMN critics $u_{\psi_1}$, $u_{\psi_2}$
$\psi_1 \leftarrow \text{ProjectHiddenWeightsToNonNegative}()$
$\psi_2 \leftarrow \text{ProjectHiddenWeightsToNonNegative}()$
**for** $i = 1$ **to** maxEpochs **do**
$\quad L_{D_{\mathrm{CT},1}}(\psi_1, \theta) = \mathbb{E}_{Y \sim \mu_0}[u_{\psi_1}(g_\theta(Y))] - \mathbb{E}_{X \sim \nu}[u_{\psi_1}(X)]$
$\quad L_{D_{\mathrm{CT},2}}(\psi_2, \theta) = \mathbb{E}_{X \sim \nu}[u_{\psi_2}(X)] - \mathbb{E}_{Y \sim \mu_0}[u_{\psi_2}(g_\theta(Y))]$
$\quad L_{d_{\mathrm{CT}}}(\psi_1, \psi_2, \theta) = L_{D_{\mathrm{CT},1}}(\psi_1, \theta) + L_{D_{\mathrm{CT},2}}(\psi_2, \theta)$
$\quad$**for** $j = 1$ **to** discriminatorEpochs **do**
$\quad\quad \psi_1 \leftarrow \psi_1 - \eta \nabla_{\psi_1} L_{D_{\mathrm{CT},1}}$ {ADAM optimizer}
$\quad\quad \psi_1 \leftarrow \text{ProjectHiddenWeightsToNonNegative}()$
$\quad\quad \psi_2 \leftarrow \psi_2 - \eta \nabla_{\psi_2} L_{D_{\mathrm{CT},2}}$ {ADAM optimizer}
$\quad\quad \psi_2 \leftarrow \text{ProjectHiddenWeightsToNonNegative}()$
$\quad$**end for**
$\quad \theta \leftarrow \theta + \eta \nabla_\theta L_{d_{\mathrm{CT}}}$ {ADAM optimizer}
**end for**
Return $g_\theta$

---

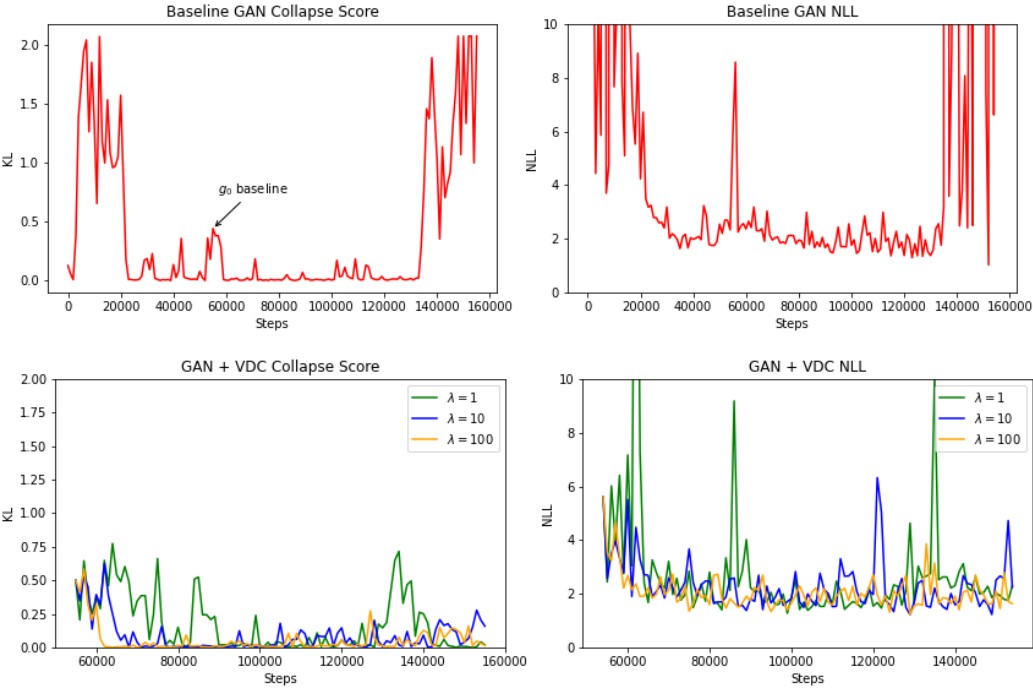

Figure 3: Probing mode collapse for GAN training.

## H ADDITIONAL EXPERIMENTAL DETAILS

**Portfolio optimization** In Figure 4 we plot the trajectory of the $z$ parameter from the portfolio optimization example solved in Sec. 7.

The convex decreasing $u$ network was parameterized with a 3-layer, fully-connected, decreasing ICMN with hidden dimension 32 and maxout kernel size of 4. See Table 2 for full architectural details. Optimization was performed on CPU.

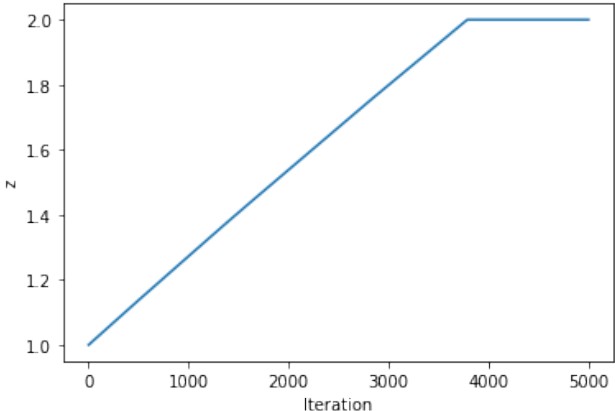

Figure 4: Trajectory of $z$ parameter from the portfolio optimization example in Sec. 7.

Table 2: Architectural details for convex decreasing $u$ network used in portfolio optimization.

| | Convex decreasing $u$ | | | |
| --- | --- | --- | --- | --- |
| | Input dimension | Output dimension | Kernel | Restriction |
| Linear | 1 | 32 | - | Non-positivity |
| MaxOut | - | - | 4 | - |
| Linear | 8 | 32 | - | Non-negativity |
| MaxOut | - | - | 4 | - |
| Linear | 8 | 1 | - | Non-negativity |

**Image generation**    In training WGAN-GP and WGAN-GP + VDC, we use residual convolutional neural network (CNN) for the generator and CNN for the discriminator (both with ReLU non-linearities). See Table 3, Table 4, and Table 5 for the full architectural details. Note that in Table 3, PixelShuffle refers to a dimension rearrangement where an input of dimension $Cr^2 \times H \times W$ is rearranged to $C \times Hr \times Wr$[3]. Our latent dimension for $\mu_0$ is 128 and $\lambda_{\mathrm{GP}} = 10$. We use ADAM optimizers (Kingma & Ba, 2015) for both networks, learning rates of $1e^{-4}$, and a batch size of 64. We use the CIFAR-10 training data and split it as 95% training and 5% validation. FID is calculated using the validation set. The generator was trained every 6th epoch, and training was executed for about 400 epochs in total. When training $g^*$ we use learning rate of $1e^{-5}$ for the Choquet critic, $\lambda = 10$, and $\lambda_{u,reg} = 10$. Training the baseline $g_0$ and $g^*$ with the surrogate VDC was done on a compute environment with 1 CPU and 1 A100 GPU.

**2D point cloud generation**    When training with the $d_{CT}$ surrogate for point cloud generation, the generator and Choquet critics are parameterized by residual maxout networks with maxout kernel size of 2. The critics are ICMNs. Our latent dimension for $\mu_0$ is 32. Both the generator and critics have hidden dimension of 32. The generator consists of 10 fully-connected layers and the critics consists of 5. For all networks, we add residual connections from input-to-hidden layers (as opposed to hidden-to-hidden). The last layer for all networks is fully-connected linear. See Table 6 for full architectural details. We use ADAM optimizers for all networks, learning rate of $5e^{-4}$ for the generator, learning rates of $1e^{-4}$ for the Choquet critics, and a batch size of 512. Training was done on a single-CPU environment.

---

[3]See `pytorch` documentation for more details.

Table 3: Architectural details for WGAN-GP generator $g_\theta$.

| WGAN Generator $g_\theta$ | | |
|---|---|---|
| | Kernel size | Output shape |
| $Y \sim \mu_0$ | - | 128 |
| ConvTranspose | $4 \times 4$ | $128 \times 4 \times 4$ |
| ResidualBlock | $3 \times 3$ | $128 \times 8 \times 8$ |
| ResidualBlock | $3 \times 3$ | $128 \times 16 \times 16$ |
| ResidualBlock | $3 \times 3$ | $128 \times 32 \times 32$ |
| BatchNorm, ReLU, Dropout | - | - |
| Conv | $3 \times 3$ | $3 \times 32 \times 32$ |
| **Residual Block** (kernel size specified above) | | |
| *Main* | LayerNorm, ReLU | |
| | PixelShuffle 2x | |
| | Conv | |
| | LayerNorm, ReLU | |
| | Conv | |
| *Residual* | PixelShuffle 2x | |
| | Conv | |

Table 4: Architectural details for WGAN-GP discriminator $f_\varphi$.

| WGAN Discriminator $f_\varphi$ | | | |
|---|---|---|---|
| | Kernel size | Stride | Output shape |
| Input | - | - | $3 \times 32 \times 32$ |
| Conv w/ReLU | $3 \times 3$ | $2 \times 2$ | $128 \times 16 \times 16$ |
| Conv w/ReLU | $3 \times 3$ | $2 \times 2$ | $256 \times 8 \times 8$ |
| Conv w/ReLU | $3 \times 3$ | $2 \times 2$ | $512 \times 4 \times 4$ |
| Linear | - | - | 1 |

Table 5: Architectural details for Choquet critic $u_\psi$ in the image domain.

| Choquet critic $u_\psi$ | | | |
|---|---|---|---|
| | Kernel size | Stride | Output shape |
| Input | - | - | $3 \times 32 \times 32$ |
| Conv | $3 \times 3$ | $2 \times 2$ | $1024 \times 16 \times 16$ |
| MaxOut + Dropout | 16 | - | $64 \times 16 \times 16$ |
| Conv | $3 \times 3$ | $2 \times 2$ | $2048 \times 8 \times 8$ |
| MaxOut + Dropout | 16 | - | $128 \times 8 \times 8$ |
| Conv | $3 \times 3$ | $2 \times 2$ | $4096 \times 4 \times 4$ |
| MaxOut + Dropout | 16 | - | $256 \times 4 \times 4$ |
| Linear | - | - | 1 |

Table 6: Architectural details for generator $g_\theta$ and Choquet critics $u_{\psi_1}, u_{\psi_2}$ in the 2D point cloud domain.

| **Generator** $g_\theta$ and **Choquet critics** $u_{\psi_1}, u_{\psi_2}$ | | |
|---|---|---|
| | Generator | Choquet critics |
| Input dim | 32 | 2 |
| Hidden dim | 32 | 32 |
| Num. Layers | 10 | 5 |
| Output dim | 2 | 1 |
| **Fully-connected Residual Blocks** | | |
| *Main* | Linear (hidden-to-hidden) MaxOut kernel size 2 Dropout | |
| *Residual* | Linear (input-to-hidden) | |

## I ASSETS

**Libraries** Our experiments rely on various open-source libraries, including `pytorch` (Paszke et al., 2019) (license: BSD) and `pytorch-lightning` (Falcon et al., 2019) (Apache 2.0).

**Code re-use** For several of our generator, discriminator, and Choquet critics, we draw inspiration and leverage code from the following public Github repositories: (1) https://github.com/caogang/wgan-gp, (2) https://github.com/ozanciga/gans-with-pytorch, and (3) https://github.com/CW-Huang/CP-Flow.

**Data** In our experiments, we use following publicly available data: (1) the CIFAR-10 (Krizhevsky & Hinton, 2009) dataset, released under the MIT license, and (2) the Github icon silhouette, which was copied from https://github.com/CW-Huang/CP-Flow/blob/main/imgs/github.png. CIFAR-10 is not known to contain personally identifiable information or offensive content.

