# OpenReview forum: "Learning with Stochastic Orders"
_ICLR.cc/2023/Conference — ICLR 2023 notable top 25%_

### Official Review · Reviewer_6Q2H · 2022-10-21

**Confidence:** 3
**Correctness:** 4
**Technical Novelty And Significance:** 4
**Empirical Novelty And Significance:** 3
**Recommendation:** 8

**Clarity, Quality, Novelty And Reproducibility:**

As indicated above, I found the paper quite clear.

Small typo in Page 3: $\mathbb E_{x \sim \mu_-} x = \mathbb E_{x \sim \mu_+} x$.


**Strength And Weaknesses:**

Overall, I found the paper well-written: while I am not an expert in stochastic orders, I could follow most of the parts.  From a theoretical viewpoint, the sample complexities about VDC and CT estimations (especially the lower bound) justify the use of ICMNs, for which the main properties are summarized in Theorems 3 and 6. The applications of this framework to multivariate portfolio selection and GAN training are relevant. Finally, the experiments on image generation look promising.

I did not find real weaknesses, but the paper is very dense. I would suggest slightly revising the introduction in order to strengthen the motivation for using convex orders in high dimensions.


**Summary Of The Paper:**

The aim of this paper is to learn probability distributions in high dimensions with dominance constraints. To this end, the authors are exploiting the Choquet order between probability distributions, and introduce the notion of Variational Dominance Criterion, a divergence measure that captures the relative spread of a distribution with respect to some baseline. Notably, the Choquet-Toland distance between two distributions can be simply expressed as a sum of symmetric variational dominance criteria. Based on these notions, the authors show that VDC suffers from the curse of dimensionality, but a VDC surrogate defined in terms of input convex max-out networks can be estimated from a polynomial number of samples, using a simple stochastic gradient descent scheme. The experimental results on portfolio optimization and image generation corroborate the merits of this approach.


**Summary Of The Review:**

In essence, it is a well-written paper about efficient learning of probability distributions under dominance constraints, with interesting applications and promising experimental results.

---

### Official Review · Reviewer_K4TL · 2022-10-22

**Confidence:** 2
**Correctness:** 4
**Technical Novelty And Significance:** 2
**Empirical Novelty And Significance:** 2
**Recommendation:** 6

**Clarity, Quality, Novelty And Reproducibility:**

Clarity: The authors define a new class order between probability densities. I am not sure how this definition relates to machine learning applications. I suggest authors motivate more on the importance of their definitions.

**Strength And Weaknesses:**

Strength: They define a new class of distance functions and order between probability densities.


Weakness:

1.  Some grammar mistakes:
"if F is the set of functions with Lipschitz" should be "if F is the set of functions with 1-Lipschitz continuity" .

2. The application of the new distance/order is not clear to me.

**Summary Of The Paper:**

The authors study the order between probability measures. They introduce a "Choquet-Toland" type distance between probability measures, which is a drop-in replacement for IPMs. They formulate a min-max framework for computing with stochastic orders. They also use the input convex maxout networks to compute the proposed distance. They demonstrate the results from numerical experiments fin high-dimensional image generation. The numerics demonstrate the successfulness of this definition. Some related analyses have been also conducted.



**Summary Of The Review:**

The paper is overall well written with clear proofs and some numerical examples. However, I am not sure about its applications, and its importance in image generation.  Some analytical examples could be important for me to judge the novelty of their definitions.

---

### Official Review · Reviewer_K5E1 · 2022-10-25

**Confidence:** 3
**Correctness:** 2
**Technical Novelty And Significance:** 2
**Empirical Novelty And Significance:** 2
**Recommendation:** 5

**Clarity, Quality, Novelty And Reproducibility:**

The reproducibility and clarity is good. The novelty might be ok. But the quality of the experiments is quite limited.

**Strength And Weaknesses:**

Strengths:

1. The motivation is clear and the written is mostly clear. The structure of the paper is good and it is easy to follow.
2. The derivation part is solid, even though I did not check the derivation in detail, from the current structure I can see that most of the derivation should be correct, and the derivation of the final surrogate function should be correct.

Weakness:

1. While the theoretic part of the paper is ok, the experimental part of the paper is kind of week. Only one experiment is done on real dataset, and the dataset is quite simple. The CIFAR10 dataset, provides images with low resolutions, and currently it would be hard to tell if the proposed approach can be applied to generate high quality images?

2. While the synthetic experiments prove that the proposed approach works well for low dimensional data, how the proposed approach would perform on high dimensional data is still very unclear based on current experimental results.

3. The proposed distance is used along with WGAN-GP in the experiments, can it work without WGAN-GP?

**Summary Of The Paper:**

The authors propose to use stochastic orders, more precisely, the Choquet order, instead of the commonly used integral probability metrics, to provide a differentiable measurement of the distance between distributions. Then this Choquet order can be applied to probability density estimation. By applying the proposed Choquet order in WGAN-GP, the approach provides slightly better FID than WGAN-GP.

**Summary Of The Review:**

The theoretic part of the paper is ok. The proposed approach is kind of more complicated than WGAN, current experiments fail to show that it is worth to apply such complicated techniques (i.e. the benefits of such techniques is still not clear).

---

### Official Review · Reviewer_QEba · 2022-10-25

**Confidence:** 3
**Correctness:** 3
**Technical Novelty And Significance:** 4
**Empirical Novelty And Significance:** 4
**Recommendation:** 8

**Clarity, Quality, Novelty And Reproducibility:**

Very good



**Strength And Weaknesses:**

**Strengths**
- the paper is globally clear and well written
- the topic addressed is of high interest to the ICLR community
- the contributions are important and well supported. In particular the surrogate VDC is very interesting. I also like that the approximation error due to the VDC surrogate is discussed at the end of Section 5
- the interest of the approach is proven empirically. Exhibiting a practical example where the dominance is needed is nice

**Weaknesses**
- from my understanding, the asymmetry of the proposed criterion is one of its main novelty and interest. Then, I feel it might be discussed more into details why the CT distance, that breaks this good feature, is interesting. In particular, how is better than standard IPMs?
- there is a notation clash between the cone $K$ and the constant in the theorems
- last line of first paragraph of Section 2: "for any $f \in \mathcal{F}$" is missing
- Thm 1: the equation numbering is weird, the push forward notation is not defined

**Summary Of The Paper:**

This paper revisits standard Optimal Transport concepts through the lens of convex orders. In particular, the authors introduce the Variational Dominance Criterion (VDC, equation 1), which possesses some interesting properties due to the asymmetry of the space ${\cal A}$ with respect to which the supremum is taken. The VDC is then thoroughly studied: statistical rates for the VDC estimation are given (Thm 2), an approximation based on maxout networks is proposed, that has parametric rate (Thm 3), a pseudo distance, the CT distance is then introduced and studied as well. Experiments are presented to show the relevance of the metrics introduced.

**Summary Of The Review:**

Overall I like the paper and think it should be accepted.

---

### Decision · Program_Chairs · 2023-01-20

**Decision:**

Accept: notable-top-25%

**Justification For Why Not Higher Score:**



**Justification For Why Not Lower Score:**



**Metareview: Summary, Strengths And Weaknesses:**

The topic of the submission is learning on the space of probability distributions in high dimension with dominance constraints. The authors introduce the Variational Dominance Criterion (VDC) in (1), and prove that it captures the Choquet order generated by convex functions (Proposition 2), relate VDC to optimal tranport (Theorem 1), and show an upper bound on its empirical measure-based estimator (Theorem 2) which indicates a curse of dimensionality. To address this bottleneck, they propose a VDC surrogate using input convex maxout networks, and prove that it possesses parametric rates (Theorem 3). They define the Choquet-Toland distance relying on VDC, show that it is indeed a distance (Theorem 4), and derive a similar conclusion as for the VDC (Theorem 5 - Theorem 6). The efficiency of the approach is illustrated in the context of portfolio optimization and image generation.

Defining novel divergence measures with efficient estimators and handling stochastic order constraints are fundamental problems of machine learning. As it was assessed by the reviewers, the authors deliver important results in this domain which can be of clear interest to the community from both theoretical and practical perspective.

**Note From Pc:**

if the above contains the word "oral" or "spotlight" please see: "oral" presentation means -> notable-top-5% and "spotlight" means -> notable-top-25%. As stated in our emails, we are disassociating presentation type from AC recommendations